

# Causes of the extensive hypoxia in the Gulf of Riga in 2018

Stella-Theresa Stoicescu[1], Jaan Laanemets[1], Taavi Liblik[1], Maris Skudra[2], Oliver Samlas[1], Inga Lips[1,3], Urmas Lips[1]

[1]Department of Marine Systems, Tallinn University of Technology, Tallinn, 19086, Estonia
5 [2]Latvian Institute of Aquatic Ecology, Riga, LV-1007, Latvia
[3]EuroGOOS AISBL, Brussels, 1000, Belgium

*Correspondence to*: Stella-Theresa Stoicescu (stella.stoicescu@taltech.ee)

**Abstract.** The Gulf of Riga is a relatively shallow bay connected to the deeper central Baltic Sea (Baltic Proper) via straits with sills. The decrease in the near-bottom oxygen levels from spring to autumn is a common feature in the gulf, but in 2018, 10 hypoxia was exceptional. We analyzed temperature, salinity, oxygen, and nutrient data collected in 2018 and historical data available from environmental databases. Forcing data from the study year were compared with their long-term means and variability. The year 2018 was exceptional due to occasionally dominating north-easterly winds supporting the inflow of saltier waters from the Baltic Proper and meteorological conditions causing fast development of thermal stratification in spring. Existing stratification hindered vertical transport between the near-bottom layer (NBL) and the water layers above it. The 15 estimated oxygen consumption rate at the sediment surface in spring-summer 2018 was about 1.7 mmol $O_2$ $m^{-2}$ $h^{-1}$ that exceeded the oxygen input to the NBL due to advection and mixing. We suggest that the observed pronounced oxygen depletion was magnified by the prolonged stratified season and haline stratification in the deep layer that maintained a decreased water volume between the seabed and the pycnocline. The observed increase in phosphate concentrations in the NBL in summer 2018 suggests a significant sediment phosphorus release in hypoxic conditions counteracting the mitigation 20 measures to combat eutrophication. We conclude, if similar meteorological conditions as in 2018 could occur more frequently in the future, such extensive hypoxia would be more common in the Gulf of Riga and other coastal basins with similar morphology and human-induced elevated input of nutrients.



# 1 Introduction

The Baltic Sea is strongly influenced by eutrophication and changing climate conditions (Conley et al., 2009; Gustafsson et al., 2012; Kabel et al., 2012). Drivers behind eutrophication are excessive amounts of nutrients, which enter the marine environment through rivers and the atmosphere. Rivers carry nutrients mainly originating from agricultural sources (e.g., fertilizers used in farming, leaching due to deforestations, etc.). Nutrients in the atmosphere are mostly from land-based sources (e.g., production of energy using fossil fuels, transport) but also from shipping (emission of nitrous oxide).

Hypoxic conditions have been found throughout the Baltic Sea as quasi-permanent, seasonal, or infrequent phenomena (Conley et al., 2007, 2011; Karlson et al., 2002). Hypoxia and anoxia occur in the open water areas of the Baltic Proper below the halocline (~70-80 m) almost permanently since the 1950s (HELCOM, 2016; Karlson et al., 2002). Oxygen conditions in the near-bottom layer of deeper areas of the central Baltic Sea, where permanent halocline exists, are occasionally improved by Major Baltic Inflows (e.g., Matthäus and Franck, 1992; Schinke and Matthäus, 1998; Schmale et al., 2016; Liblik et al., 2018). In the Gulf of Finland, the south-westerly wind forcing could cause the reversal of estuarine circulation, leading to the collapse of stratification in the cold season and subsequent oxygenation of sub-halocline layers (Liblik et al., 2013; Lips et al., 2017).

There are also shallower regions, where halocline is absent but seasonal thermocline restricts vertical mixing and oxygen consumption leads to temporal hypoxia in the near-bottom layer in late summer-autumn, increasing the release of sediment phosphorus (Lukkari et al., 2009; Puttonen et al., 2014, 2016; Walve et al., 2018). For instance, such seasonal hypoxic events occur in the northern Baltic coastal areas and Åland archipelago, influenced by large-scale eutrophication driven by nutrients from agriculture and local fish farms (Bonsdorff et al., 1996). Human-induced elevated inputs of nutrients can cause severe oxygen deficiency events in the case of certain meteorological/hydrographic conditions, as it happened in 1994 and 2002 in the southern Baltic and the Danish coastal waters (e.g. Conley et al., 2007; Powilleit and Kube, 1999).

One of the shallow areas, where seasonal hypoxia occasionally occurs, is the Gulf of Riga (GoR) in the eastern part of the Baltic Sea (e.g., Berzinsh, 1995 and references therein; Aigars and Carman, 2001; Eglīte et al., 2014; Aigars et al., 2015). The Gulf of Riga is a semi-enclosed shallow basin (Fig. 1) with a surface area of 16,330 $km^2$, a volume of 424 $km^3$, and a mean depth of 26 m (Ojaveer, 1995; HELCOM, 2002). The gulf's deeper central area situated east of the Ruhnu island has depths up to 56 m (Stiebrins and Väling, 1996). Water and salt budget of the gulf if governed by river discharge, precipitation-evaporation and water exchange with the Baltic Proper through the connecting straits. The long-term (1950–2015) mean river runoff is about 36 $km^3$ $year^{-1}$ (Johansson, 2016) and the average freshwater flux due to the difference between the precipitation and evaporation rates is about 2.5 $km^3$ $year^{-1}$ (Omstedt et al., 1997). Five larger rivers (Daugava, Lielupe, Gauja, Pärnu, and Salaca) enter the southern and eastern part of the gulf whereas the Daugava river contributes about 70% to the total riverine





input (Yurkovskis et al., 1993). Assuming the gulf's water volume and salt content's annual balance, Lilover et al. (1998) estimated the water renewal period about three years.

GoR water exchange with the Baltic Proper takes place via the Irbe Strait in the west (about 70-80% of water exchange) and

the Suur Strait in the north (Petrov, 1979). The Irbe Strait has a sill depth of 25 m and a cross-section area of 0.4 km$^2$, while the Suur Strait 5 m and 0.04 km$^2$, respectively. Lips et al. (1995) suggested that the gulf deep layer water could be renewed in summer by inflows of saltier water from the eastern Baltic Proper over the sill in the Irbe Strait, which is deeper and wider, while inflows through the shallow Suur Strait are arrested in the surface layer. The near-bottom inflows through the Irbe Strait are intensified by the northerly and north-easterly winds that cause upwelling events along the eastern coast of the Baltic

Proper. Model simulations by Raudsepp and Elken (1995) also showed that strong northerly wind events could create substantial near-bottom inflows of saltier Baltic Proper waters. However, when downwelling occurs along the eastern coast of the Baltic Proper, the inflowing water is warmer than the near-bottom layer water in the Gulf of Riga in summer, and it can spread buoyantly at the intermediate depths (Liblik et al., 2017). A general cyclonic circulation in the Gulf of Riga with the southward flow on the western side and northward flow on the eastern side of the gulf was described by Yurkovskis et al.

(1993). In accordance with this flow scheme, freshwater from the south-eastern part of the gulf moves toward the north along the eastern shore in the surface layer while the inflowing saltier water through the Irbe Strait moves toward the south as a geostrophically balanced gravity current along the gulf's western slope. The numerical study by Lips et al. (2016) suggested a more complicated (seasonally altered) whole-basin circulation in the surface layer depending on the prevailing wind forcing and stratification.

Because of the shallowness of the basin, the whole water column is well mixed and vertical distributions of temperature, salinity and oxygen are homogenous in winter. In summer, stratification is mainly maintained by the seasonal thermocline, which starts to develop in April and is the strongest in August, and the contribution of haline stratification is rather moderate (Stipa et al., 1999; Liblik et al., 2017). Skudra and Lips (2017) revealed based on summer CTD profiles from 1993–2012 that

the strongest stratification of water column occurred in the years with the highest upper layer temperature in summer and river runoff in spring. High correlation between the deep layer salinity in the Irbe Strait and the gulf was found by Skudra and Lips, (2017) in accordance with the suggestion that the majority of water exchange between the Baltic Proper and the gulf occurs through the Irbe Strait.

The general annual cycle of dissolved oxygen (DO) concentration in the Gulf of Riga could be described as follows (Berzinsh, 1995 and references therein): in the surface layer, oxygen concentrations are close to the saturation in winter, maximum concentrations are observed during the vernal phytoplankton bloom in late April and early May, and a decrease in the surface layer oxygen concentrations appears due to the decay of the spring bloom and increasing water temperature in summer; oxygen conditions in the near-bottom layer are influenced by the amount of decomposing organic material, lateral transport, and mixing





in the water column, the lowest concentrations are observed in late summer and autumn and the maximum close to saturation in winter when the water column is completely mixed.

Based on data from 1963 to 1990, a statistically significant decreasing trend of oxygen concentration in August was found for
the entire 20–50 m layer in the gulf (Berzinsh, 1995). No trend was detected after that (HELCOM, 2009). The latest monitoring data are not analyzed for long-term trends and inter-annual variations in near-bottom oxygen concentrations; instead, model outcomes are used to describe the oxygen conditions (e.g., Jansson et al., 2020). However, it is well documented that the anoxic and hypoxic areas have been expanding in the entire Baltic Sea during the last decades due to both eutrophication and the changes in climatic conditions (Hansson and Viktorsson, 2020; the analysis also includes data from the Gulf of Riga). For a
northern Baltic coastal basin, it has been suggested that in addition to the anthropogenic nutrient input the shoaling of the basin and warming contributed to the deoxygenation of bottom waters (Jokinen et al., 2018). Otherwise, it was impossible to explain more frequent seasonal hypoxia occurrences since the beginning of the 1900s.

Total annual nitrogen and phosphorus loads to the Gulf of Riga estimated for 2017 at levels of 90 544 and 2 427 t year$^{-1}$,
respectively, are still higher than the maximum allowable inputs according to the Baltic Sea Action Plan, and almost no trends in inputs have been observed compared to the reference period 1997–2003 (HELCOM, 2019). Based on monitoring data since 1974, the phosphorus pool in the Gulf of Riga constantly increased till the mid-1990s (Yurkovskis, 2004); afterward, a tendency has been unclear (HELCOM, 2018b). Since the phosphorus delivery by rivers is <15% compared to its pool (Yurkovskis, 2004), the changes in the latter are largely governed by internal processes. The phosphate flux from the sediments
to the water column depends on the near-bottom oxygen conditions with maximum values at low DO concentrations. For instance, values in the order of 100 µmol $PO_4^{3-}$ m$^{-2}$ d$^{-1}$ were simulated at oxygen concentrations 1-2 mg l$^{-1}$ (Eglīte et al., 2014). Thus, the reoccurrence of poor near-bottom oxygen conditions supports sediment phosphorus release that counteracts potential decreases in the external phosphorus load to the gulf.

The Gulf of Riga experienced extensive near-bottom hypoxia in summer–autumn 2018. This study aims to evaluate the possible role of different forcing factors leading to the observed hypoxia. For that, we analyzed the dissolved oxygen, stratification and forcing data in 2018 and compared these with the long-term means and variability. A more detailed comparison with the conditions in 2017, when hypoxia did not develop, was conducted. In addition, we estimated the oxygen consumption rates and sediment phosphorus release under the observed hypoxic conditions.

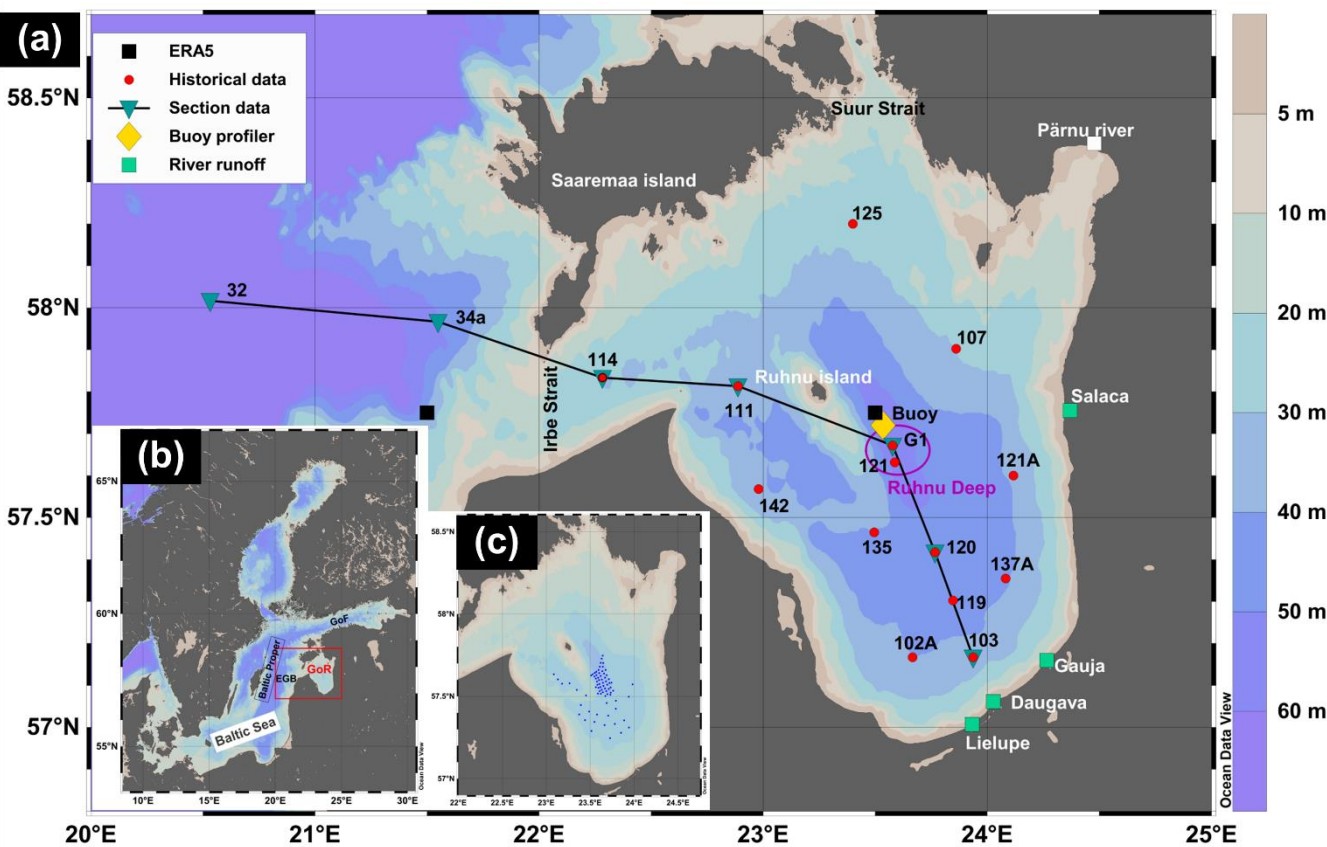

Figure 1. (a) - Map of the study area in the Gulf of Riga (GoR) with bottom topography. Red filled circles represent the locations of monitoring stations. Yellow filled diamond represents the location of buoy profiler. Inverted triangles represent the stations used for section figures. Black filled squares represent the grid cell center points of ERA5 data (grid cell resolution 0.25° x 0.25°). Green filled squares mark rivers from where runoff data were used. EGB – Eastern Gotland Basin, GoF – Gulf of Finland. (b) – Study area in the Baltic Sea. (c) – Survey stations in September, 2018. This map was generated using Ocean Data View 5.2.0 software (Schlitzer, 2019) (Schlitzer, Reiner, Ocean Data View, odv.awi.de, 2019).



## 2 Material and methods

Data sets from various sources covering different periods with different sampling/recording frequencies were used. Vertical profiles of temperature, salinity, and dissolved oxygen were recorded using an Ocean Seven 320plus CTD probe (Idronaut s.r.l.) onboard r/v Salme during Estonian and Latvian monitoring cruises in 2012–2018 (stations are shown in Fig. 1). Salinity

and density anomaly are shown in the present study as Absolute Salinity (g kg$^{-1}$) and Sigma-0 (kg m$^{-3}$) and were calculated using the TEOS-10 formula (IOC et al., 2010). The oxygen sensor (Idronaut s.r.l.) attached to the OS320plus probe was calibrated before each cruise. Oxygen profiles used for the analysis were quality checked against the laboratory analysis of water samples using an OX 400 l DO (WWR International, LCC) analyzer. The accuracy of the Idronaut oxygen sensor is 0.1 mg l$^{-1}$, while the accuracy of the laboratory dissolved oxygen analyzer is 0.5 % of the measured value. The primary data set,

from where the vertical profiles and sections were drawn and oxygen consumption estimated, was collected within the following six campaigns in 2018: 9-10 January, 17-18 April, 30 May, 11 July, 25 August and 26-27 October.

Nutrient concentrations were determined from water samples collected onboard r/v Salme during monitoring cruises in 2018 using the automatic nutrient analyzer Lachat QuikChem 8500 Series 2 (Lachat Instruments, Hach Company). The nutrient

analyses were performed according to the recommendations by USEPA, ISO, and DIN standards (methods 31-107-04-1-D $NO_3$ (Egan, 2000) and 31-115-01-1-I $PO_4$ (Ammerman, 2001)). The lower detection range for $PO_4^{3-}$ and $NO_2^- + NO_3^-$ was 0.06 and 0.08 μM, respectively.

High-frequency measurements were carried out with a buoy profiler in the gulf's deepest part in summer 2018 (Fig. 1). The

profiler system Mona (Flydog Solutions Ltd.) recorded vertical profiles of temperature, salinity, dissolved oxygen, and chlorophyll-a fluorescence eight times a day from 5 to 21 August. It includes an OS316plus CTD probe (Idronaut s.r.l.) with an Idronaut oxygen sensor and Trilux fluorescence sensor (Chelsea Technologies Group Ltd.). The profiler records data at a rate of 8–9 Hz while moving down with an average speed of 8–10 cm s$^{-1}$. The accuracy of the used Idronaut oxygen sensor is 0.1 mg l$^{-1}$. After the pre-processing of data, the vertical profiles were stored with a constant step of 0.5 m from 3 m to 50 m.

Data from the research vessel cruises were used for the quality assurance of the measurements by the buoy profiler.

Historical data on temperature, salinity, dissolved oxygen, and nutrient concentrations were obtained from the Estonian environmental monitoring information system (KESE), Latvian environmental monitoring databases, ICES/HELCOM database, and SeaDataNet Pan-European infrastructure for ocean and marine data management (http://www.seadatanet.org)

for the period 2005–2018.

Meteorological data for two locations were extracted from the ERA5 dataset via Copernicus Services (see in Fig.1; a grid cell has horizontal dimensions of 0.25º×0.25º) (Hersbach et al., 2018). Hourly net solar radiation and wind data at the 10 m height



from the grid cell centered at 57º30′ N, 23º30′ E in the gulf from 1979 to 2018 were obtained to characterize local conditions. Surface net solar radiation (J m⁻²) is the amount of solar radiation (shortwave radiation) reaching the surface of the Earth (both direct and diffuse) minus the amount reflected by the Earth's surface. Also, hourly wind data at the 10 m height from the grid cell centered at 57º 30′ N, 21º 30′ E (outside the gulf) for March–August 2017 and 2018 were extracted to find the periods

with upwelling-favorable conditions along the eastern coast of the Baltic Proper.

River runoff data (1993–2018) were received from the Latvian Environment, Geology and Meteorology Center and contain the estimated monthly runoff (m³ s⁻¹) of rivers Salaca, Gauja, Lielupe, and Daugava (Fig 1). River runoff is presented as monthly average flow (m³ s⁻¹) per river.

Oxygen concentration $\leq 2.9$ mg l⁻¹ was used as the threshold concentration for hypoxia, and the upper boundary of the hypoxic layer was found as the minimum depth where oxygen concentration was below the threshold. The estimated depth of the upper boundary of the hypoxic layer at station G1 (see Fig. 1) and the gridded topography (EMODnet Bathymetry Consortium, 2020) were used to find the lateral extent of the hypoxic area. In 2018, special surveys were conducted in the gulf's central deeper

area, and the area of hypoxic near-bottom water was also estimated using the survey mean depth of the hypoxic layer upper boundary.

The depth of the upper mixed layer (UML) was defined according to Liblik and Lips (2012) as the minimum depth, where $\rho_z - \rho_3 > 0.25$ kg m⁻³, where $\rho_z$ is the density anomaly at depth z and $\rho_3$ at depth 3 meters. The depth of the near-bottom mixed

layer (NBL) was found similarly to UML, as the maximum depth, where $|\rho_z - \rho_{last}| > 0.1$ kg m⁻³, where $\rho_{last}$ is the density anomaly at the maximum depth of a profile.

We introduce a rough method estimating oxygen consumption rates in the gulf NBL. If neglecting mixing between the NBL and the water column above it, salinity changes in the NBL should be attributed to the lateral advection and mixing of the NBL

water with the inflowing waters. Knowing salinities of inflowing waters and gulf NBL waters at time steps *t1* ("old" water) and *t2* ("new" water), we can estimate the proportion of inflowing waters in the near-bottom water mass at time step *t2*. Using this proportion, we can also estimate the expected changes in the NBL oxygen concentration due to lateral transport and mixing. The expected oxygen concentration of the "new" water can be found as

$$O_2^{t2}(G1) = O_2^{t1}(G1) + \left(O_2^{t1}(114) - O_2^{t1}(G1)\right) * \left[\frac{Sal^{t2}(G1) - Sal^{t1}(G1)}{Sal^{t1}(114) - Sal^{t1}(G1)}\right], \tag{1}$$

where the parameters of the "old" water are defined as salinity and oxygen concentration at an initial time step *t1* in the NBL at station G1 ($Sal^{t1}(G1)$ and $O_2^{t1}(G1)$, respectively) and the inflowing water as salinity and oxygen concentration in the NBL at station 114 in the Irbe Strait ($Sal^{t1}(114)$ and $O_2^{t1}(114)$, respectively; see Fig. 1). $Sal^{t2}(G1)$ is salinity of the "new" water defined as salinity in the NBL at station G1 at time step *t2*.





To account for the changes in the NBL salinity and oxygen concentration due to vertical mixing between the NBL and upper water layers, we used a similar approach as by Stoicescu et al. (2019). Diffusive flux through the border between the NBL and the water column above can be estimated based on the vertical gradient of salinity $DIFF_S = k * {\partial S}/{\partial z}$ or oxygen $DIFF_{O_2} =$

$k * {\partial O_2}/{\partial z}$, where the vertical diffusivity coefficient is calculated as $k = {\alpha}/{N}$, $\alpha$ is the empirical intensity factor of turbulence (we applied a constant value $\alpha = 1.5 * 10^{-7}$ $m^2$ $s^{-2}$) and $N$ is the Brunt-Väisälä frequency defined by the vertical density gradient. The changes in salinity and oxygen concentration can be found by multiplying the values of diffusive fluxes with the time between two measurements ($t2-t1$) and dividing with the thickness of the NBL ($h_{NBL}$). As a consequence, salinity at station G1 at time step t2 used in Eq. 1 should be found by subtracting from the measured salinity value ($Sal^{t2m}(G1)$) the

estimated change due to diffusion as $Sal^{t2}(G1) = Sal^{t2m}(G1) - DIFF_S * (t2 - t1)/h_{BNL}$.

Due to oxygen consumption, measured oxygen concentration in the NBL at station G1 at time step $t2$ ($O_2^{t2m}(G1)$) should be lower than that found when considering only physical processes (lateral advection and mixing, estimated by Eq. 1 and vertical diffusion) since no production is expected in the near-bottom layer that is well below the euphotic depth. Dividing the

difference between the expected and measured oxygen concentration by the time between two measurements ($t2-t1$), we can estimate the oxygen depletion rate due to consumption. Oxygen consumption per unit bottom area is calculated as

$$O_2^{consumption} = \left(O_2^{t2}(G1) - O_2^{t2m}(G1)\right) / (t_2 - t_1) * h_{NBL} + DIFF_{O_2}. \tag{2}$$

We have chosen the time step of one month or longer to estimate oxygen consumption rates based on the distance between the Irbe Strait and the Ruhnu Deep (120 km, measured along the deeper area of the gulf) and average (monthly) flow rates in the

gulf of 5 cm $s^{-1}$ (e.g., Soosaar et al., 2014; Lips et al., 2016). The applicability of the introduced consumption rate estimates is more thoroughly analyzed in the Discussion section.

The same approach of using the changes in salinity for defining the proportion of water masses in the mixture was used to estimate the flux of phosphates from the sediments. In Eqs. (1) and (2), oxygen concentration was replaced by phosphate

concentration. The difference in measured and expected phosphate concentrations in the gulf near-bottom layer was associated with the phosphorus release from the sediments.



# 3 Results

## 3.1. Inter-annual and seasonal variability

### 3.1.1. Inter-annual and seasonal changes in near-bottom oxygen and phosphate concentrations

Data at the central, deep stations G1 and 121 (see Fig. 1; further on, these stations are treated as one station) in 2005–2018
showed high variability of oxygen concentration in the near-bottom layer (Fig. 2). High oxygen concentrations were measured
in winter when the gulf's water column was well mixed. Seasonal hypoxia in the near-bottom layer was occasionally registered
in several years from August to November. No hypoxia was observed in 2006–2011 (except one value close to 2.9 mg l$^{-1}$ in
2009), but note the scarce sampling frequency. Using the series of the deepest, simultaneously measured salinity and oxygen
content values at station G1 from August to November in 2005-2018, we found a statistically significant ($p < 0.05$, $R^2 = 0.21$,
$n = 35$) negative relationship – low oxygen values correspond to high salinity values. However, there were examples when
hypoxia occurred at salinities slightly above 5.5 g kg$^{-1}$ (in 2015) and did not exist at salinities 6.5 g kg$^{-1}$ (in 2010).

We estimated temporal trends in the average summer (August) and autumn (October-November) near-bottom oxygen and
phosphate concentrations using data from 2005-2018 at all stations with depth >= 40 m (see station locations in Fig. 1). A
statistically significant ($p < 0.05$) decreasing trend in dissolved oxygen at a rate of 0.45 mg l$^{-1}$ year$^{-1}$ was found in autumn ($n$
$= 13$, $R^2 = 0.50$). Statistically significant increasing trends were found for the near-bottom phosphate concentration in summer
(0.08 µM year$^{-1}$, $n = 14$, $R^2 = 0.47$) and autumn (0.12 µM year$^{-1}$, $n = 13$, $R^2 = 0.34$). As expected from these results, statistically
significant negative correlation was obtained between the autumn near-bottom layer mean oxygen and phosphate
concentrations ($n = 13$, $R^2 = 0.79$). The autumn near-bottom layer oxygen also significantly correlated with the next year winter
(January) near-bottom layer phosphate concentration ($n = 9$, $R^2 = 0.59$).





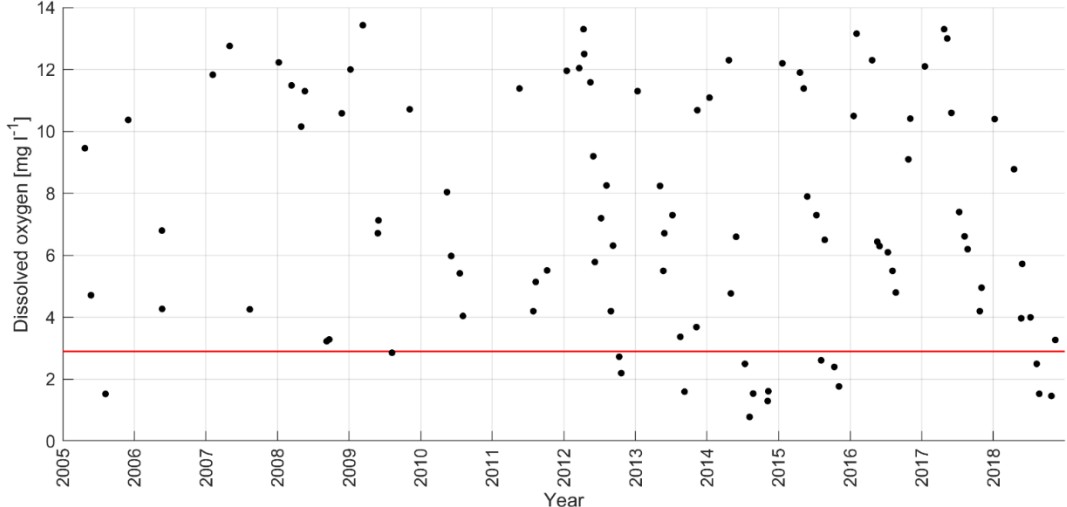

Figure 2. Inter-annual variability of near-bottom dissolved oxygen concentration using the deepest measured values at monitoring stations G1 and 121 in 2005–2018. A bold red line marks the oxygen concentration 2.9 mg l$^{-1}$ (threshold concentration for hypoxia).

### 3.1.2. Inter-annual and seasonal variability in vertical distribution of temperature, salinity, density, and oxygen

Time series of the vertical distributions of temperature, salinity, density anomaly, and oxygen concentration at monitoring station G1 in 2012–2019 are presented in Fig. 3. A clear seasonal course was observed in all parameters. During the cold season, when the water column was practically mixed, the lowest temperatures and largest oxygen concentrations were 10 observed (Fig. 3a and d). Inter-annual changes in the summer (June-September) UML temperature and depth as well as vertical stratification were large, most probably due to the variation in atmospheric forcing (surface heat flux and wind stress). UML was deeper in the summers 2012 and 2017, and mean surface layer temperatures were higher (23.5 °C) in 2014 and (22.4 °C) in 2018 (Fig. 3a). Lower mean surface layer salinities (5.2-5.3 g kg$^{-1}$) were observed in 2012, 2013, 2014 and 2018 (Fig. 3b). Increased salinity in the near-bottom layer was observed in the summers 2013 (6.3-6.5 g kg$^{-1}$) and 2018 (6.5-6.8 g kg$^{-1}$) (Fig. 15 3b), indicating saline water inflows from the Baltic Proper into the gulf through the Irbe Strait. The low salinity in the surface layer and the saline water inflows contributed to the strongest water column stratification in these years (Fig. 3c) – the density difference between the bottom and surface layer was 2.6 kg m$^{-3}$ and 2.8 kg m$^{-3}$ in 2013 and 2018, respectively. The weakest stratification was in 2012 and 2017 when the respective density difference was 1.5 kg m$^{-3}$.

Figure 3. Time series of the vertical distribution of temperature (a), salinity (b), density anomaly (c), and oxygen concentration (d) at stations G1 and 121 in 2012-2019. Vertical white dashed lines mark the time of measured profiles.

The lowest oxygen concentrations, below the hypoxia threshold, were observed in the near-bottom layer in summer/autumn 2012 (2.2 mg l$^{-1}$), 2014 (0.8 mg l$^{-1}$), 2015 (1.8 mg l$^{-1}$) and 2018 (1.5 mg l$^{-1}$) (Fig. 3d). The bottom area covered by hypoxic waters, estimated using the profiles measured at station G1, was up to 4.4% (703 km$^2$) in 2012, 2.4% (382 km$^2$) in 2014, and 2.1% (338 km$^2$) in 2015. The estimated extent of hypoxia was the largest in 2018 when the hypoxic water covered 5.2% (834 km$^2$) of the gulf's bottom area.

The hypoxic area was also estimated using the survey data from September 2018 instead of a single profile (see panel C in Fig. 1). Profiles with a minimum depth of $\geq 45$ m were selected (n = 50). The mean depth, where DO concentration was 2.9 mg l$^{-1}$, was 44.5 m, corresponding to a hypoxic area of 935 km$^2$ (5.9%). Hypoxia was detected in 92% of the selected profiles. Hypoxic depth values varied from the mean with a standard error of 2.1 m, and the coefficient of variation was < 4 %. These estimates show that the survey data gave the estimate of the hypoxic area close to the estimate using only the oxygen profile at a central station. Vertical profiles at station G1 acquired in August and October both gave the estimate of 834 km$^2$ while the survey in September 935 km$^2$. The latter points to a maximum hypoxia extent in September, between the regular environmental monitoring cruises in August and October. Using the dense survey data, we can get confidence limits for hypoxic depth estimates from a single profile considering the detected ca 4 % variation of the results.

## 3.2 Analysis of long-term forcing data

### 3.2.1 Wind conditions

We compared wind conditions in 2018 with the long-term mean wind conditions by calculating monthly mean wind vectors for 1979–2018 and 2018 from ERA5 wind data (Fig. 4a). For this analysis, the data point located in the EGB (see Fig. 1) was used. Mean wind vectors in February, March, May, July and September in 2018 differ considerably from the long-term mean wind vectors. North-easterly winds supporting the inflow of saline water from the Baltic Proper through the Irbe Strait into the gulf prevailed in February-March, May and July 2018. In September 2018, south-westerly winds prevailed, but the magnitude was larger than usual. To characterize wind-induced mixing, we also compared the monthly average wind speed in 2018 with the wind speed statistics for the period 1979-2018 (Fig. 4b). Here, the data point located in the Gulf of Riga (see Fig. 1) was used. The monthly average wind speed from February to August was lower in 2018 than the average for the respective month, except in April. The lowest wind speed for the entire period 1979-2018 was found in May 2018, suggesting that the wind-induced mixing of the water column in May was the weakest in 2018 if compared with the other years.

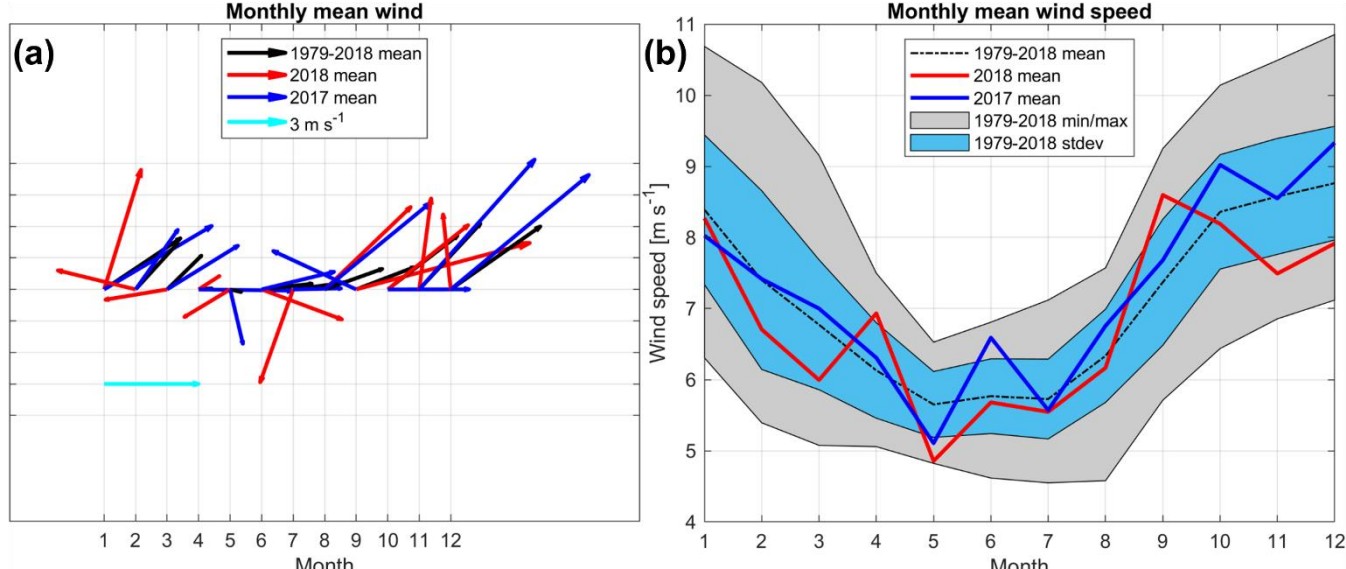

Figure 4. (a) Monthly mean wind vectors in 1979–2018 (black arrows), 2017 (blue arrows) and 2018 (red arrows) in the EGB close to the Irbe Strait. (b) Monthly mean wind speed with standard deviation and minimum-maximum values of monthly mean wind speed in 1979-2018 and monthly mean wind speed in 2017 and 2018 in the central Gulf of Riga (see locations of grid cells in Fig. 1).

To find similar years to the year 2018 in terms of prevailing north-easterly winds, we analyzed monthly mean wind vectors in 2005-2017, when we have regular dissolved oxygen measurements in the Gulf of Riga. Although the winds were predominantly from the south-westerly direction in 1979-2018, northerly (from north-west to north-east) winds were observed for 1-2 months in spring-summer for several years in 2005-2018. These years and months were 2005 (March – NNE, April and June - WNW), 2006 (March – NE, July - NW), 2008 (April – NE, May - N), 2009 (April – WNW, June – NNW), 2010 (May – NNE), 2011 (April – NW), 2012 (March - NW), 2013 (March – NE, July- NW), 2014 (May and June – NW) and 2015 (April - WNW). The years with no dominating winds from northerly direction were 2007, 2016 and 2017. Low oxygen concentrations in the near-bottom layer of the central gulf were observed in the late summer-autumn of 2005, 2012-2015 and 2018 (Fig. 2). Thus, the anomalous winds in spring-early summer are a characteristic feature for the years when low oxygen concentrations occurred in late summer-autumn, but such winds alone were not sufficient for causing the hypoxia.

### 3.2.2 River runoff

Analysis of monthly mean river runoff data (sum of rivers Salaca, Gauja, Lielupe, and Daugava) showed considerable variability, especially in spring (March–May) (Fig. 5). A comparison of monthly river runoff values from the year 2018 with the long-term mean values (1993–2018) shows that runoff was mostly lower than the long-term mean, although within standard





deviation limits or not exceeding previous minimum-maximum values. An exception was January 2018, when runoff was the largest on the record, being more than twice as large (4.90 km$^3$ month$^{-1}$) as the long-term mean (2.25 km$^3$ month$^{-1}$). The maximum or close to the maximum value of monthly river runoff were also observed in September-December 2017. Thus, a larger than average river runoff in autumn-early winter 2017-2018 could cause slightly lower salinity in the gulf in spring

2018. However, a lower than average runoff in spring-summer 2018 could not strengthen the vertical stratification of the water column in late summer-autumn.

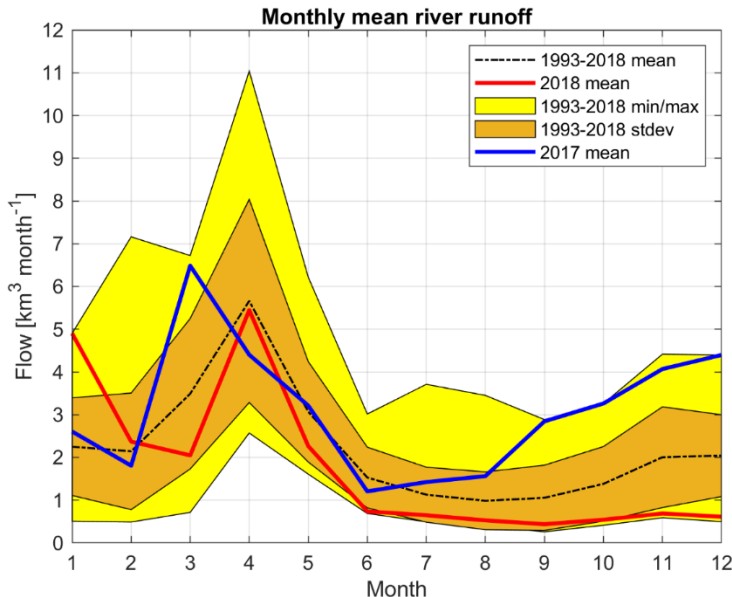

Figure 5. The course of mean monthly river runoff (dashed line), monthly minimum/maximum runoff (yellow area) and

standard deviation (orange area) for 1993–2018. Bold red and blue lines mark monthly mean river runoff for the years 2018 and 2017, respectively.

### 3.2.3 Surface net solar radiation and air temperature

Based on ERA5 hourly surface net solar radiation data, we calculated the monthly mean net solar radiation and air temperature

in 1979–2018 for the grid point close to station G1 (see Fig. 1 for location). The monthly mean net solar radiation in May 2018 was the highest during the observation period (Fig. 6a), and in June 2018, it was the third highest value after 1992 and 1979. Seasonal variation in air temperature in 2018 differed from the average (Fig. 6b). While air temperature in 2018 was lower than the average in February-March, a rapid increase of air temperature occurred in April-May, and it stayed higher than the monthly averages in 1979–2018 until October (Fig. 6b). Also, monthly mean air temperature in September 2018 was the

highest of the observation period, and July and August 2018 were in the top 5 warmest years.





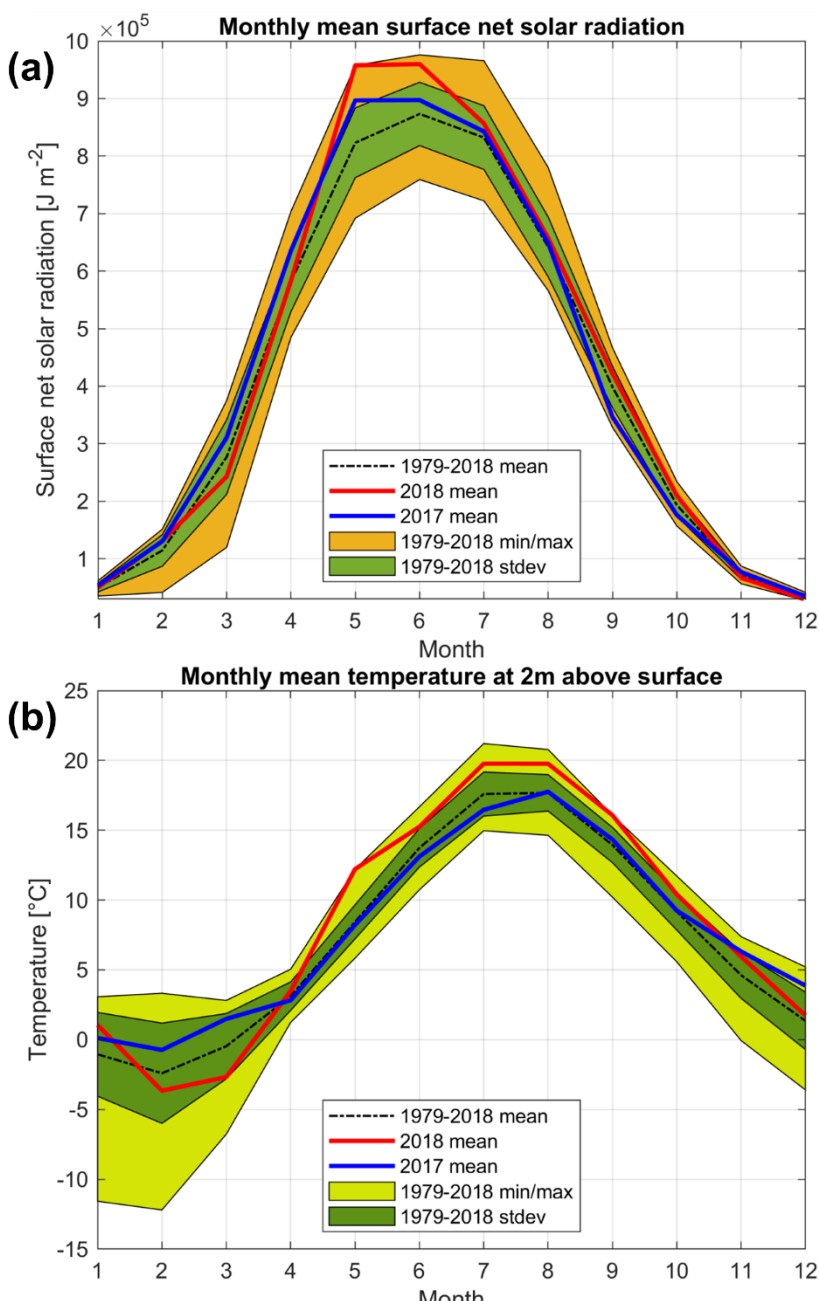

Figure 6. (a) Course of monthly mean surface net solar radiation (dashed line), monthly minimum/maximum (orange area), and standard deviation (green area) of radiation for the period 1979–2018 (ERA5 data). The bold red and blue lines mark monthly mean radiation for the year 2018 and 2017, respectively. (b) Course of monthly mean air temperature at 2 meters above surface (dashed line), monthly minimum/maximum (light green area), and standard deviation (darker green area) of temperature for the period 1979–2018 (ERA5 data). The bold red and blue lines mark monthly mean temperature for the year 2018 and 2017, respectively.





### 3.3. Deep layer dynamics and stratification

#### 3.3.1. Oxygen, salinity and temperature sections through the Irbe Strait in 2018

For a qualitative description of probable links of the changes in the deep layer with the water exchange over the sill in the Irbe Strait, vertical sections of oxygen, salinity and temperature from the eastern Baltic Proper to the Gulf of Riga from May to

5 August 2018 were constructed (Fig. 7). Based on the measurements at the end of May, the intrusion of saline water has reached Station 111 (near-bottom salinity was > 6.8 g kg$^{-1}$), while salinity was > 6.0 g kg$^{-1}$ (up to 6.2 g kg$^{-1}$) in the near-bottom layer of deep areas of the gulf. Higher temperatures were associated with the saltier water in the near-bottom layer at Station 111, indicating that these waters could result from the mixing of gulf waters with the warmer surface waters from the Baltic Proper. Oxygen concentrations in the near-bottom layer in the central gulf and towards the Daugava river mouth in the southern gulf

were at a level of 60% of saturation. Low temperatures of this near-bottom water mass suggest that it has been formed in winter or early spring as a mixture of local Gulf of Riga winter waters and saltier waters originating from the Baltic Proper. Higher salinity and lower oxygen content towards the Daugava river than in the Ruhnu Deep at the same depth could be related to the inclination of the boundary between the near-bottom and deep water due to atmospheric forcing and probably to more intense oxygen consumption at the sediment surface close to the mouths of major rivers.

In July, the deep area around monitoring station G1 was filled with higher salinity waters, > 6.4 g kg$^{-1}$, and oxygen concentrations had decreased to 40% of saturation (< 5 mg l$^{-1}$). Salinity in the near-bottom layer in the Irbe Strait was as high as in late May, but saltier waters' intrusion towards the central gulf was not so intense as during the previous survey. The measurements in late August suggest that the further filling of deeper areas with saline water continued between the surveys –

20 salinity increased up to 6.55 g kg$^{-1}$ at monitoring station G1. Hypoxic conditions with oxygen saturation below 20% were observed in the near-bottom layer of the central gulf. However, the salinity, temperature and oxygen distributions in the deep layers near the Irbe Strait area indicate that most probably outflow from the gulf prevailed below the seasonal thermocline at the end of August 2018.





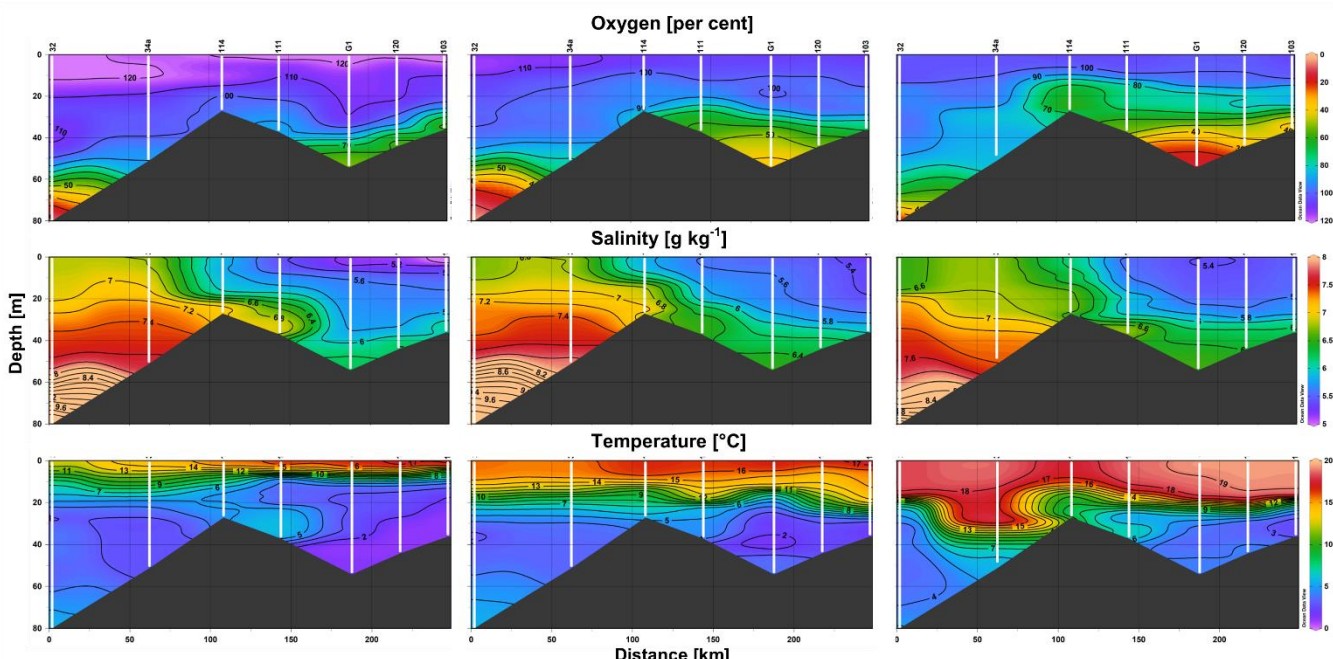

Figure 7. Vertical sections of oxygen saturation (upper panels), salinity (center panels) and temperature (lower panels) on 30 May (left panels), 11 July (center panels) and 25-26 August (right panels) 2018 along the route from station 32 in the Baltic Proper through the Irbe Strait to station 103 in the Gulf of Riga (see the locations in Fig. 1).

### 3.3.2. Comparative analysis of vertical distributions and wind forcing in 2017 and 2018

As described above, the most extensive hypoxia since 2005 was observed in the near-bottom layer of the central Gulf of Riga in 2018. The below analysis of the observed changes in vertical distributions and wind forcing was conducted, bearing in mind a suggestion that the 2018 hypoxia was likely related to the spreading of saline waters through the Irbe Strait (Fig. 7) that

10 created vertical stratification in the deep layer. To highlight the hypoxia-favorable conditions in 2018, the data from 2018 are compared with the respective data from 2017 that we chose as an example year when hypoxia did not develop. Also, as noted above, in 2017, monthly average winds did not display northerly winds that would support inflows.

Vertical profiles of temperature, salinity, density, and dissolved oxygen concentration in the Ruhnu Deep demonstrated

15 considerable differences between the years 2017 and 2018 (Fig. 8). By the end of May-early June 2017, the upper mixed layer was thick and relatively cold (9.7 °C), while in 2018, a thin and warm (up to 18.0 °C) near-surface layer formed (Fig. 8a and e). Though, near-bottom layer temperatures were <5 °C in both years. The surface layer was fresher in spring 2018 – salinity varied in April–May/June between 5.44 and 5.86 g kg$^{-1}$ in 2017 and 5.14 and 5.65 g kg$^{-1}$ in 2018. In 2017, near-bottom salinity variations were small, within the range of 5.90–6.06 g kg$^{-1}$, while in 2018, near-bottom salinity was higher and the changes



were larger than in 2017. Relatively high salinity was observed in the near-bottom layer already in April and salinity increased almost continuously from April to July 2018 from 6.06 to 6.50 g kg$^{-1}$, indicating the spreading of saltier waters into the Ruhnu Deep during that period.

The differences in vertical distributions of temperature and salinity in 2017 and 2018 were reflected in the water column stratification (Fig. 8c and g). The water column was well mixed at the end of April 2017. A single pycnocline with changing strength and thickness was observed between the upper mixed layer and the well-mixed deep layer throughout the summer 2017. In 2018, the situation was different – a pycnocline between the upper less saline water and saltier near-bottom water was observed below the 40 m depth in April. By the end of May 2018, two pycnoclines were formed, the deep pycnocline located
below 35 m depth and the upper pycnocline (seasonal thermocline) at the 10 m depth. The intermediate layer (10–35 m) remained cold and well mixed. A relatively thin, practically mixed near-bottom layer was observed from May to October. The largest bottom to surface density difference was observed in August in both years – 2.05 kg m$^{-3}$ in 2017 and 2.76 kg m$^{-3}$ in 2018. Thus, vertical stratification was stronger in 2018.

Temporal courses of near-bottom oxygen concentrations in the Ruhnu Deep (Fig. 8d and h) showed clear decreasing trends from spring to autumn in 2017 and 2018. In 2018, oxygen concentration decreased simultaneously with the increase of NBL salinity until mid-July. From mid-July to the end of August, near-bottom salinity increased only slightly but hypoxia developed in the near-bottom layer. The signs of vertical mixing, as an expansion of the near-bottom layer and a decrease in salinity, are apparent when comparing the vertical profiles from August and October 2018. However, hypoxic conditions with no decrease
in oxygen concentration prevailed until the end of October 2018. In summer-autumn 2017, the NBL salinity only slightly increased, and hypoxia was not observed.

A prominent feature in the Ruhnu Deep in 2018 was the haline stratification in the deep layer, potentially restricting mixing between the near-bottom layer and the water column above. The spreading of saltier waters from the eastern Baltic Proper to
the Gulf of Riga over the Irbe Strait sill was demonstrated by the vertical sections of salinity (Fig. 7). The time series of along-coast component (NNE–SSW) of wind stress revealed the periods in both years, 2017 and 2018, when the upwelling-favorable winds with negative wind stress $\tau_N$ exceeding –0.2 N m$^{-2}$ occurred, supporting the inflows into the near-bottom layer of the central gulf (Fig. 9). However, the main difference between these two years is expressed by the cumulative wind stress, which followed the long-term pattern in 2017 but clearly deviated from it in 2018. From February to the end of July in 2018, wind
forcing, on average, supported the inflow of saltier waters into the Gulf of Riga, as revealed by the decrease in the cumulative wind stress (Fig. 9). From the beginning of August, the winds from opposite direction, which could cause downwelling along the eastern coast of the Baltic Proper, prevailed. Thus, the inflows of sub-thermocline waters into the Gulf of Riga could have been blocked in late summer 2018.



Figure 8. Vertical profiles of temperature (a, e), salinity (b, f), density anomaly (c, g), and dissolved oxygen concentration (d, h) measured in the Ruhnu Deep (station G1, see location in Fig. 1) in 2017 (left panels) and 2018 (right panels).



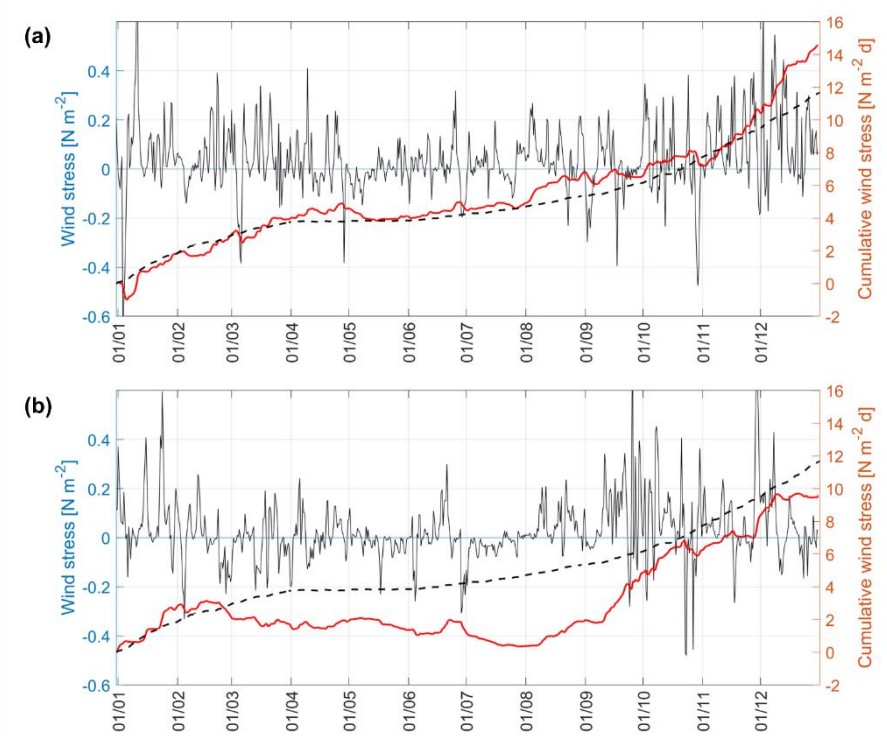

Figure 9. Time series of local along-coast (NNE–SSW) component of wind stress $\tau_N$ (black line, positive northward), cumulative wind stress (red line) in 2017 (a) and 2018 (b) (6h moving average is shown) and cumulative wind stress based on mean wind stress for 1979-2018 from 1 January to 31 December. See the location of the ERA5 grid point in the EGB close to the Irbe Strait in Fig. 1.

## 3.4. High-frequency dynamics of temperature, salinity, and dissolved oxygen in relation to the wind forcing

High-frequency measurements with the buoy profiler were carried out from 5 to 21 August 2018 close to station G1 (see Fig. 1). Time series of the vertical distributions of temperature, salinity and oxygen together with the wind speed variability are presented in Fig. 10. Two strong wind events passed the Gulf of Riga on 10 and 12 August, when the average wind speed was up to 11.3 and 13.6 m s$^{-1}$, respectively. The first wind pulse slightly changed the upper mixed layer depth. Although the thickness of the hypoxic, saltier layer slightly increased after the first wind event, the variability of salinity and oxygen concentration in the near-bottom layer was small until the second wind event. The second wind pulse on 12 August caused a rapid deepening of the thermocline and an increase of the upper mixed layer depth by about 8 m. Near-bottom layer salinity and thickness increased during the wind pulse. At the same time, the oxygen concentration remained approximately on the same level, but the thickness of the hypoxic layer increased about 8 m. The wind speed decreased on 13 August and remained low (< 5 m s$^{-1}$) until 17 August. The decrease of wind speed was accompanied by the decrease of salinity and increase of



oxygen concentration over the hypoxia threshold in the near-bottom layer until the noon of 15 August. A rapid increase of salinity and decrease of oxygen concentration occurred in the second half of 15 August. A relatively thick hypoxic layer remained at the measurement site until 18 August. Further, until the end of measurements, the decreasing trend of hypoxic layer thickness and salinity was observed, gradually approaching conditions similar to the beginning of the measurements. To

conclude, the buoy profiler measurements displayed high variability of the upper mixed layer temperature and depth caused by wind-induced mixing.

High variability of near-bottom layer salinity and oxygen concentration and thickness of the near-bottom haline hypoxic layer, in the time scale of few days, was most probably caused by the wind forced advection, not vertical mixing between the NBL

and the water layer above it. Deepest measured oxygen concentrations varied from 1.3 to 3.3 mg l-1, with the mean of 2.1 mg l-1, when taking into account profiles with the maximum depth of $\geq$ 50 m. The standard deviation of the deepest oxygen measurements was 0.50 mg l-1, and the coefficient of variation 24 %. These statistical parameters give an estimate of the uncertainty in using near-bottom oxygen values from a single measurement that should represent a two-week period (when the monitoring is conducted with a time step of two weeks). The uncertainty is higher for the current monitoring program with a

sampling step of 1.5-2 months.

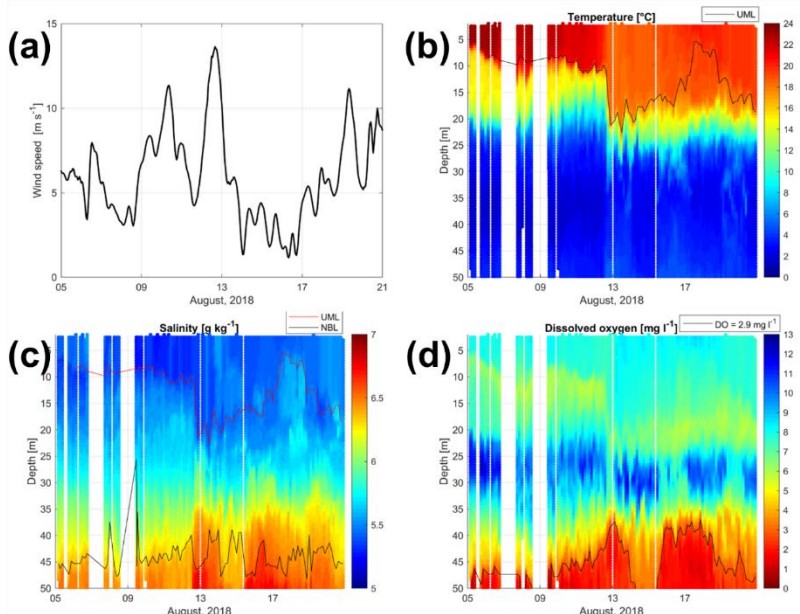

Figure 10. Time series of wind speed (ERA 5; 5h moving average) at the location close to the buoy station (a) and vertical distribution of (b) temperature, (c) salinity and (d) dissolved oxygen measured at the buoy station from 5 to 21 August 2018.

DO = 2.9 mg l-1 marks the hypoxia threshold. UML is the upper mixed layer and NBL is the near-bottom mixed layer, both defined as described in Sect. 2.





### 3.5. Estimates of oxygen consumption and sediment release of phosphates

Oxygen consumption was estimated using Eqs. 1 and 2, based on the changes in salinity and oxygen concentration in the near-bottom mixed layer in the Ruhnu Deep and the Irbe Strait. A comparison of the measured salinity changes and estimated changes in the NBL due to diffusion shows that the salt flux due to advection and lateral mixing had about three times larger contribution to the salinity changes than vertical diffusion (Table 1; row "$Sal^{t2}(G1) - Sal^{t1}(G1)$ due to diffusion (g kg$^{-1}$)") from mid-April to mid-July while it was only slightly larger from mid-July until the end of August 2018. Both advection and vertical diffusion should increase the oxygen concentration in the gulf NBL and had comparable contributions varying from 0.67 mg l$^{-1}$ to 1.74 mg l$^{-1}$ and 0.75 mg l$^{-1}$ to 1.46 mg l$^{-1}$, respectively, for studied three periods (Table 1). Advection and lateral mixing had the largest influence on the oxygen content when the difference between the NBL oxygen concentrations between the central gulf and the Irbe Strait was the largest. However, the measurements and calculations according to the introduced approach show that oxygen consumption had to be large enough to exhaust oxygen brought by advection and diffusion and cause further oxygen depletion in the NBL. The estimates of the consumption rate per unit bottom area, which could be a more relevant indicator for oxygen consumption as opposed to oxygen depletion per volume unit of the NBL, assuming that oxygen consumption takes mostly place on the sediment surface, varied from 1.53 to 1.75 mmol m$^{-2}$ h$^{-1}$ in 2018 (Table 1; row "DO consumption rate per unit bottom area").

We also assessed consumption rates from August to October in 2018, but the prevailing conditions (outflow from GoR through the Irbe Strait) were unsuitable for applying the method. We have assumed an inflow of oxygenated EGB sub-thermocline waters through the Irbe Strait to deeper areas of the GoR. Secondly, a stable stratification of the water column, which hinders vertical mixing between the NBL and the water layer above, should exist to apply the gradient method of diffusion flux estimates. For instance, in spring 2017, the near-bottom layer was not separated by a vertical density gradient from the layers above. Thus, the gradient method of estimating vertical salt and oxygen flux is not applicable. Also, oxygen concentrations in the NBL of the Irbe Strait were similar or even lower than the oxygen concentration at station G1 in 2017, probably due to prevailing outflows of deep GoR waters through the Irbe Strait.





Table 1. Estimated changes in NBL salinity and oxygen concentration due to advection and diffusion and estimated consumption rates between the monitoring campaigns in 2018. Parameters measured in or estimated for monitoring campaigns' start/end are denoted by t1/t2. Sal marks salinity, DO marks dissolved oxygen and NBL marks near-bottom mixed layer.

| Period start (t1) | 18.04.2018 | 30.05.2018 | 11.07.2018 |
|---|---|---|---|
| Period end (t2) | 30.05.2018 | 11.07.2018 | 25.08.2018 |
| $Sal^{t1}(114)$ (g kg$^{-1}$) | 6.77 | 7.17 | 7.27 |
| $Sal^{t1}(G1)$ (g kg$^{-1}$) | 5.99 | 6.22 | 6.48 |
| $Sal^{t2}(G1)$ (g kg$^{-1}$) | 6.22 | 6.48 | 6.51 |
| $Sal^{t2}(G1) - Sal^{t1}(G1)$ due to diffusion (g kg$^{-1}$) | -0.14 | -0.13 | -0.11 |
| $O_2^{t1}(114)$ (mg l$^{-1}$) | 13.42 | 12.46 | 11.08 |
| $O_2^{t1}(G1)$ (mg l$^{-1}$) | 12.01 | 8.27 | 5.16 |
| $O_2^{t2m}(G1)$ (measured, mg l$^{-1}$) | 8.27 | 5.16 | 2.52 |
| $O_2^{t2}(G1)$ estimated due to advection and lateral mixing (mg l$^{-1}$) | 12.68 | 10.01 | 6.18 |
| $O_2^{t2}(G1) - O_2^{t1}(G1)$ due to advection and lateral mixing (mg l$^{-1}$) | 0.67 | 1.74 | 1.02 |
| $O_2^{t2}(G1) - O_2^{t1}(G1)$ due to diffusion (mg l$^{-1}$) | 1.46 | 1.09 | 0.75 |
| NBL thickness (m) | 9.5 | 9.5 | 12.0 |
| **DO consumption rate per unit bottom area (mmol O$_2$ m$^{-2}$ h$^{-1}$)** | **1.72** | **1.75** | **1.53** |

In 2018, hypoxic conditions were observed at station G1 and with the developing oxygen depletion, phosphate concentrations increased in the near-bottom layer (Fig. 11). However, note that the phosphate concentrations were already elevated in July when the near-bottom oxygen values (2-3 m from the seabed) did not indicate hypoxic conditions there yet. Still, hypoxia could be the case at the sediment-water interface. The amount of phosphates released from sediments was estimated using the

same approach as for oxygen consumption (see Sect. 2), taking into account the concentrations in the near-bottom layer at stations G1 and 114 – the latter representing the inflowing water. Since the vertical resolution of sampling was scarce (step was 10 m), we used only the deepest measured phosphate concentration as the value characterizing the entire NBL, and the vertical gradient was estimated between phosphate concentrations at 50-52 m and 40 m.

The changes in phosphate concentrations due to advection and lateral mixing between the "old" near-bottom water and "new" inflowing water and vertical diffusion were estimated using Eqs. 1 and 2 and the gradient method. As seen from Table 2, the changes due to advection and lateral mixing were larger than due to vertical diffusion. Considering both physical processes, the concentration estimates were lower than the measured values on the next monitoring campaign. This difference between the measured and estimated concentrations was assigned to the sediment release of phosphates. The largest estimated

phosphate flux from the sediments was 13.6 µmol m$^{-2}$ h$^{-1}$ for the period from the end of May to mid-July. From late April to





late May, the phosphate flux from the sediments was minimal, which might be explained by relatively high oxygen concentrations in the NBL. Although the oxygen concentrations decreased and fell below the hypoxia threshold from mid-July to late August, the estimated sediment release was lower for this period (13.6 μmol m$^{-2}$ h$^{-1}$) than the preceding period (from late May to mid-July).

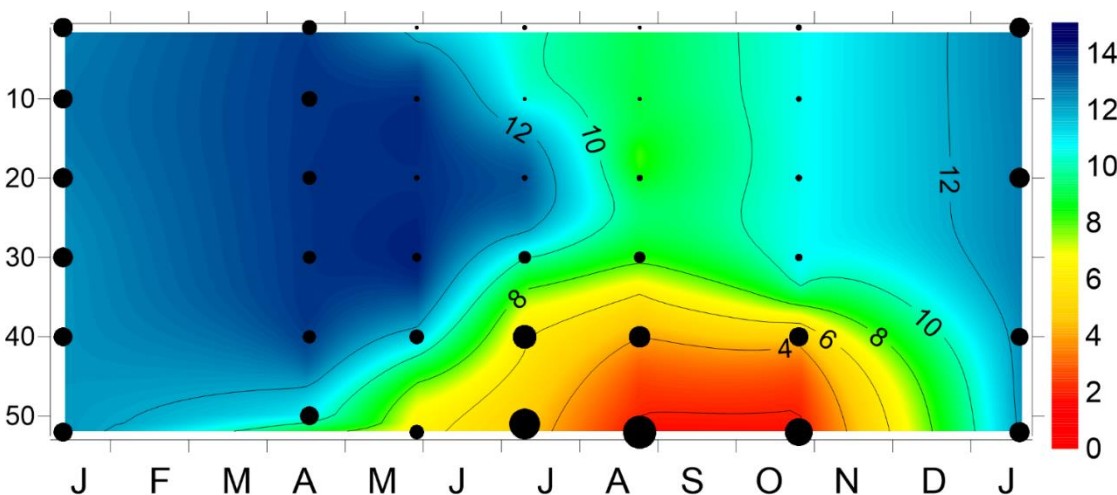

Figure 11. Time series of vertical distribution of oxygen and phosphate concentration from January 2018 to January 2019. Phosphate concentrations are indicated as black dots. The size of dots is proportional to the measured concentrations ranging from the lower detection range of 0.06 to 2.25 μM.





Table 2. Estimated changes in NBL phosphate concentration due to advection and diffusion and estimated sediment release of phosphates in the Gulf of Riga between the monitoring campaigns in 2018. Parameters measured in or estimated for monitoring campaigns' start/end are denoted by t1/t2.

| | | | |
|---|---|---|---|
| Period start (t1) | 18.04.2018 | 30.05.2018 | 11.07.2018 |
| Period end (t2) | 30.05.2018 | 11.07.2018 | 25.08.2018 |
| $PO_4^{t1}(114)$ (μM) | 0.49 | 0.21 | 0.33 |
| $PO_4^{t1}(G1)$ (μM) | 1.12 | 0.82 | 1.95 |
| $PO_4^{t2m}(G1)$ (measured, μM) | 0.82 | 1.95 | 2.11 |
| $PO_4^{t2}(G1)$ estimated due to advection and lateral mixing (μM) | 0.84 | 0.57 | 1.63 |
| $PO_4^{t2}(G1) - PO_4^{t1}(G1)$ due to advection and lateral mixing (μM) | -0.28 | -0.25 | -0.32 |
| $PO_4^{t2}(G1) - PO_4^{t1}(G1)$ due to diffusion (μM) | -0.06 | -0.07 | -0.19 |
| NBL thickness (m) | 9.5 | 9.5 | 12.0 |
| **Estimated rate of the sediment PO₄ release (μM m⁻² h⁻¹)** | **0.4** | **13.6** | **7.4** |



## 4 Discussion

The Gulf of Riga, a relatively shallow basin with limited water exchange with the open Baltic Sea and considerable nutrient loads from rivers, is strongly affected by eutrophication (HELCOM, 2018b). Based on data from 1963 to 1990, a statistically

significant dissolved oxygen decrease in August was found for the entire 20-50 m layer in the gulf (Berzinsh, 1995). From the late 1980s to 2006, the means of the lowest 25% of oxygen concentrations measured 1 m above the seabed in autumn ranged between 1.8 and 4.8 ml l$^{-1}$ (2.6 and 6.9 mg l$^{-1}$) without an apparent trend (HELCOM, 2009). The present study points to the decreasing trend in summer–autumn oxygen concentrations since 2005 in the Gulf of Riga near-bottom layer and occasional hypoxia events. We found seven out of 14 years (2005–2018) when the near-bottom oxygen concentration during summer–

autumn was <2.9 mg l$^{-1}$. In the years, when hypoxia occurred, seasonal stratification was typically enhanced by the vertical salinity distribution, often with higher salinity in the near-bottom layer. In 2018, fast warming of the surface layer in spring and spreading of saltier waters into the near-bottom layer in the first half of the year resulted in strong vertical stratification and extensive hypoxic area in the gulf in August–October. Thus, it seems that hypoxic conditions develop in the near-bottom layer if stratification is strengthened, hindering vertical mixing. However, the factors contributing to the occurrence and

severity of near-bottom hypoxia have to be analyzed in more detail since the inflow of saltier waters, if originating from the near-surface layer, should feed the near-bottom layer of an enclosed basin with oxygen (e.g. Schmidt et al., 2021).

We suggest that, in addition to the enhanced water column stratification, certain other factors or processes, and the timing, how and when the vertical stratification developed, contributed to the observed extensive hypoxia in 2018. Conditions,

favoring larger than usual oxygen depletion, were pre-set already by exceptionally large river runoff at the end of 2017 and the beginning of 2018 since larger river runoffs during autumn-winter raise the nutrient and organic matter concentrations in the gulf (Yurkovskis, 2004). Not fully mixed water column in the gulf deep area after the winter season 2018 could be the other important factor. In late spring, the gulf' water column is usually homogenous (Skudra and Lips, 2017; Stipa et al., 1999), but in April 2018, we observed vertical salinity and dissolved oxygen gradients in the deep layer. The appearance of the saltier

near-bottom layer matches with the observed easterly winds in February-March 2018 which could create the inflows of Baltic Proper waters through the Irbe Strait (Lips et al., 1995; Raudsepp and Elken, 1995).

A detailed comparison of oxygen dynamics between the years 2018 and 2017 indicated that the largest difference was a drastic drop of near-bottom oxygen concentrations in April-May 2018. It could be related to the existing haline stratification in the

deep layer and, thus, restricted vertical mixing, while dissolved oxygen consumption was intensified due to the settling of the vernal bloom (Olli and Heiskanen, 1999). High heat flux through the sea surface and weakest winds in May 2018 compared to all other years caused strong thermal stratification and weak vertical mixing. In 2017, such a decrease in oxygen concentrations did not occur (although we could assume a similar oxygen consumption due to the settling spring bloom) since





the water column was not stratified yet, and dissolved oxygen could be transported to the near-bottom layer from the upper water layers. Furthermore, the near-bottom mixed (weakly stratified) layer extended from the seabed to the depths of 38.5 m to 30.5 m in May-August 2017, while its thickness did not exceed 12 m in 2018 (from the seabed to 42 m depth). It could be a reason for more pronounced oxygen depletion in 2018 than in 2017. This effect of stronger depletion in 2018 is similar to the suggestion by Jokinen et al. (2018) that a decrease of the water volume between the pycnocline and the seabed increases the probability of hypoxia occurrences.

From mid-July to the end of August 2018, salinity in the near-bottom layer did not increase considerably anymore, but oxygen concentrations dropped below the hypoxia threshold. Thus, almost no saltier water inflows into the near-bottom layer existed in late summer 2018, and lateral advection and mixing could not supply additional oxygen there. It agrees with the prevailing downwelling-favorable winds along the EGB coast near the Irbe Strait since the beginning of August, which did not support the inflows anymore. The observed changes in the upper mixed layer depth also could contribute to the lack of inflows. As seen from the buoy profiler data, strong winds caused a deepening of the upper mixed layer in mid-August 2018. Such sharp or gradual deepening of the upper mixed layer, expected during summer months (from June to August) in the Gulf of Riga and similar basins (Liblik and Lips, 2011; Stipa et al., 1999), also influences the characteristics of the inflow water. If even the inflows occur, the waters could be too warm (since originating from the warm upper layer or thermocline), and as described by Liblik et al. (2017), such inflows could occur as buoyant sub-surface intrusions. As a result, the near-bottom layer of the gulf would not receive additional oxygen through lateral transport. We conclude that since mid-August, due to both missing inflow favorable winds and deepening of the thermocline, lateral transport to the deepest part of the gulf was almost absent, and hypoxia developed there.

A way to describe the level of eutrophication is to estimate the amount of oxygen consumed by organic material degradation (Carstensen et al., 2014; Stoicescu et al., 2019). Because the inflowing saltier Baltic Proper waters have higher oxygen concentration than oxygen concentrations in the central Gulf of Riga near-bottom layer, hypoxia has to be locally developed. We estimated oxygen consumption rates for the central gulf using an indirect method by assuming that the changes in the near-bottom oxygen concentrations were mostly driven by advection and lateral mixing (inflow of saltier waters through the Irbe Strait), vertical diffusion and local consumption. Vertical mixing is incorporated in the introduced method considering a complete mixing in the limits of the NBL and diffusive flux between the NBL and upper layers defined by the vertical gradient and turbulent diffusion coefficient depending on the strength of the vertical stratification (e.g., Stoicescu et al. (2019)). Note that this method of estimating diffusion is applicable only in the case of stable stratification and vertical gradients, not varying too much. For instance, in the case of complete mixing of the water column, it could not be applied. We got an average consumption rate estimate of 1.67 mmol $O_2$ $m^{-2}$ $h^{-1}$ from mid-April to late August 2018.



The reliability of the introduced method is also related to the chosen time step and the use of single profiles acquired once a month or more seldom. If considering an average water speed of 5 cm s$^{-1}$ (e.g. Lips et al., 2016; Soosaar et al., 2014) and the distance from the Irbe Strait to the Ruhnu Deep of 120 km, the 'traveling' duration of an inflowing water mass would be about 28 days. Thus, the monitoring campaigns apart 1-1.5 months could be acceptable for applying this method since the time step

5   between the measurements is long enough. To assess the potential error of using infrequent profiles (although sampled at almost regular dates), we found consumption rates based on high-frequency profiler data with up to 8 profiles a day for about two weeks on 5-21 August 2018 (Fig. 12). Monitoring profiles from late May 2018 were selected as the starting point for the estimations. Based on profiler data, we found the mean consumption rate value of 1.53 mmol $O_2$ m$^{-2}$ h$^{-1}$. The standard deviation of the estimates was 0.37 mmol $O_2$ m$^{-2}$ h$^{-1}$ and coefficient of variation 25 %.

On the one hand, it was a surprise that the results were relatively stable since, as shown above, the variability in the near-bottom layer was quite high on 5-21 August 2018. On the other hand, it was expected since this variability could mostly be assigned to the advection and changes in the thickness of the near-bottom layer at the measurement site, but not to the mixing with the water column above the NBL. The mean consumption rate estimated using buoy profiler data (1.53 mmol $O_2$ m$^{-2}$ h$^{-1}$)

15   is almost the same as the average for 2018 based on monitoring campaigns from late May to late August (1.64 mmol $O_2$ m$^{-2}$ h$^{-1}$). We conclude that the method can be used even based on scarce monitoring profiles, but one must remember the uncertainties. In our case, the standard error was 0.37 mmol $O_2$ m$^{-2}$ h$^{-1}$ and relatively large variability of the estimates even between the profiles collected during the same day can be noticed (Fig. 12). In extreme cases, the estimates could differ up to 2 times (from <1 to >2 mmol $O_2$ m$^{-2}$ h$^{-1}$) that makes using single profiles rather unreliable and should encourage to increase

20   the monitoring frequency and the use of profilers (Mack et al., 2020).



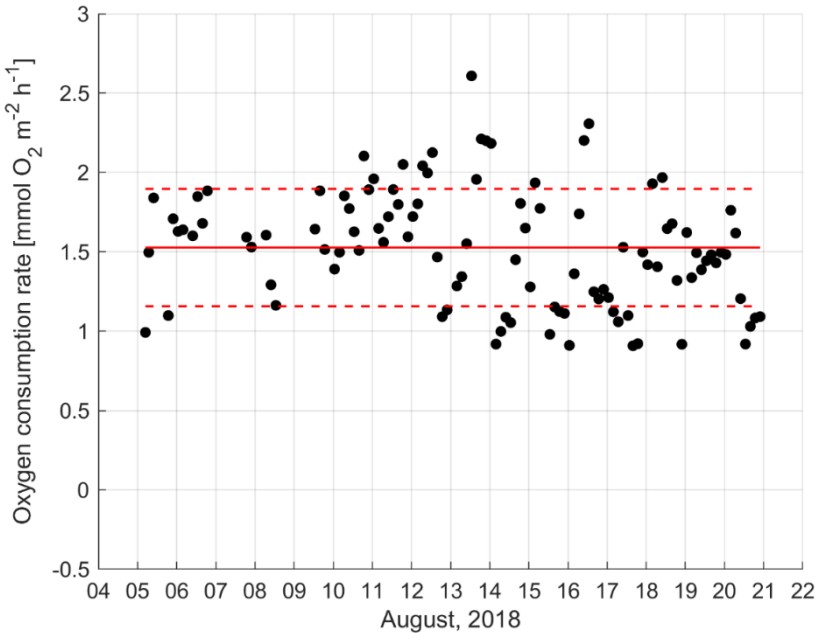

Figure 12. Oxygen consumption rates estimated using buoy profiler data from 3-18 August 2018 and monitoring data on 30 May 2018 as the salinity and dissolved oxygen values of the "old" waters (station G1) and inflowing waters (station 114). Red solid line represents the mean oxygen consumption rate of 1.53 mmol $O_2$ m$^{-2}$ h$^{-1}$. Red dashed lines represent the standard deviation bounds.

Our average consumption rate estimates for the spring-summer of 2018 – based on monitoring campaigns from late April to late August (1.67 mmol $O_2$ m$^{-2}$ h$^{-1}$) and monitoring and profiler data from late May to August (1.53 mmol $O_2$ m$^{-2}$ h$^{-1}$) – are comparable with the earlier measurements and calculations. Results from this study are higher than the estimates of consumption rates obtained for the Baltic Proper and the Gulf of Finland – 0.11-0.39 mmol $O_2$ m$^{-2}$ h$^{-1}$ (Koop et al., 1990) and 0.46-0.53 mmol $O_2$ m$^{-2}$ h$^{-1}$ (Conley et al., 1997), respectively, and closer to but slightly lower than the estimates based on the direct measurements in the Gulf of Riga by Aigars et al. (2015) – on average 2.3 mmol $O_2$ m$^{-2}$ h$^{-1}$. One reason of lower values obtained in our study might be related to the fact that the analysis by Aigars et al. (2015) was conducted using core samples collected at stations 119 and 120, which are situated in the southern Gulf of Riga close to the main river mouths (see Fig. 1).

Possible future changes in climate, including warming that causes the strengthening of stratification and decreased oxygen solubility, and changes in precipitation/river runoff, influence the extent of hypoxia in the Baltic Sea (Meier et al., 2011). Jokinen et al. (2018) have suggested that in addition to the changes in anthropogenic nutrient loads and climate change, the long-term alteration in morphology, leading to increased source-to-sink ratio and the decrease of the bottom water volume,





could increase the vulnerability to hypoxia in enclosed basins (separated from the open sea areas by sills). We add that basin-specific relationships between the basin morphology (e.g., sill depth) and stratification parameters (e.g., UML depth) should be considered. If the sill depth is less than the thermocline depth, the inflows could not reach the near-bottom layer in the deeper parts of the enclosed basins like the inflows into the Gulf of Riga via the Suur Strait. If the depths of the thermocline
and the sill in the connecting strait are comparable, then due to an average deepening of the thermocline during the summer months (Liblik and Lips, 2011), such conditions of no inflows into the NBL of deeper areas occur more often in late summer-autumn. In the very long run, the shoaling of the sills in the northern Baltic Sea due to glacio-isostatic uplift (with a rate of up to 4 mm y$^{-1}$; e.g., Mäkinen and Saaranen, 1998) could be a reason for more frequent hypoxia today than hundreds years ago.

The peak of the spring bloom, which generates most of the sedimented organic material, is observed in the Gulf of Riga in April-May (Olli and Heiskanen, 1999; Purina et al., 2018). When the spring bloom material reaches the sediment surface, it triggers enhanced oxygen consumption. Aigars et al. (2015) found higher consumption rates in late spring-early summer than in late summer-autumn and related this result to the availability of organic material, i.e., settling of spring bloom. Our results with high consumption rates from May to mid-July 2018 agree with their estimates and interpretation. Thus, the strength of
stratification in spring, including its early development, is a crucial factor influencing the extent of seasonal hypoxia in the Gulf of Riga and similar coastal basins. In general, a longer duration of the stratified season, including earlier onset of stratification in spring and later decay of vertical stratification in autumn, will lead to a more extended vegetation period (Wasmund et al., 2019) and hypoxia presence in the near-bottom layer of seasonally stratified basins. The latter could have caused the observed sharp decline in near-bottom oxygen concentrations in the Gulf of Riga in autumn during the last 15 years.

Although phosphorus inputs into the Gulf of Riga have decreased (HELCOM, 2018c), they are still higher than maximum allowable inputs by about 1000 t year$^{-1}$ (HELCOM, 2019). Our analysis of the long-term nutrient data revealed a statistically significant increasing trend in near-bottom phosphate and total phosphorus concentrations in the stratified season. Earlier studies (e.g., Yurkovskis, 2004) and assessments (HELCOM, 2018a, 2018d) have revealed that the trends in both total
phosphorus and phosphate concentrations in the surface layer (as used for the referred HELCOM indicators) are unclear since the 1990s. The increase in near-bottom phosphate concentrations could be attributed to the changes in internal load – the release of phosphates from the bottom sediments under low oxygen concentrations (e.g. Pitkänen et al., 2001). The resuspension-induced flux (e.g. Almroth et al., 2009) could have a notable contribution in the Gulf of Riga, as short-term high values of vertical shear of horizontal velocity forced by local strong wind events have been reported by Raudsepp and Kõuts
(2001) in the Ruhnu Deep. However, the analyzed long-term data support the suggestion on the increased release of phosphates in conditions of more extended hypoxia in recent years – the inter-annual variability in phosphate and oxygen concentrations in summer-autumn are well correlated. We observed an increase in phosphate concentrations in the near-bottom layer in summer 2018 already before hypoxia development. It agrees with the results obtained by Aigars et al. (2015) that the phosphate flux did increase substantially when the near-bottom oxygen concentrations fell below 6 mg l$^{-1}$.



We estimated phosphate fluxes using an indirect method with an assumption that the near-bottom phosphate concentrations change due to the internal load, advection and lateral mixing of inflowing waters with the gulf near-bottom waters and vertical diffusion. The fluxes estimated with a time step of approximately 1.5 months reached up to 13.5 µmol m$^{-2}$ h$^{-1}$ (end of May to mid-July 2018). Earlier studies by Eglīte et al. (2014) and Aigars et al. (2015) have obtained similar results. They measured the changes in nutrient concentrations during 48-hour cycles of laboratory incubation of sediments under controlled oxygen concentrations varying from 1 to 10 mg l$^{-1}$ (Eglīte et al., 2014) or corresponding to the near-bottom oxygen concentrations at the sampling site and time (Aigars et al., 2015). The average phosphate fluxes directed out of the sediments obtained by Eglīte et al. (2014) varied between 4.8 µmol m$^{-2}$ h$^{-1}$ at low oxygen concentrations (1-2 mg l$^{-1}$) and 1.8 µmol m$^{-2}$ h$^{-1}$ at 10 mg l$^{-1}$. Aigars et al. (2015) found that the average phosphate fluxes from sediments gradually increased from low values (2-5 µmol m$^{-2}$ h$^{-1}$) in April-May to 10 µmol m$^{-2}$ h$^{-1}$ in June-July, 18 µmol m$^{-2}$ h$^{-1}$ in August and 55 µmol m$^{-2}$ h$^{-1}$ in October 2012. They suggested that inorganic fractions of phosphorus (and nitrogen) mineralized from organic material are retained in sediments until near-bottom water oxygen concentration is substantially decreased. Our maximum flux estimates from late May to mid-July are close to the values obtained by Aigars et al. (2015) in June-August but we did not observe a further increase of NBL phosphate concentrations in late summer-autumn and the flux estimates from mid-July to late August were lower than for the preceding period. The sediment flux of phosphates obtained in this study also agrees with an average estimate for the coastal Gulf of Finland – 13 kg km$^{-2}$ d$^{-1}$ or 17 µmol m$^{-2}$ h$^{-1}$ (Pitkänen et al., 2001).



## 5 Conclusions

We suggest that the sequence of certain processes triggered the observed extensive hypoxia in the Gulf of Riga in 2018. First, haline stratification was formed in the deep layer of the central gulf in early spring, most probably due to inflows of saltier waters forced by the prevailing easterly winds in February-March 2018. Enhanced seasonal stratification was created by the rapid warming of the surface layer and calm wind conditions in spring leading to restricted vertical mixing. Thus, oxygen depletion due to consumption at the sediment surface was confined in the relatively thin near-bottom mixed layer already in spring when usually the water column is weakly stratified and oxygen could be supplied to the deep areas by vertical mixing. North-easterly winds that occasionally dominated in spring-summer 2018 supported the inflows of saltier waters through the Irbe Strait that maintained haline stratification in the deep layer of the gulf. In August-October, no further inflows reached the near-bottom layer of the central gulf due to unfavorable wind conditions and deepening of the thermocline, resulting in a drop of oxygen concentrations below the hypoxia threshold. Assuming that oxygen depletion in the near-bottom layer was governed by consumption at the sediment surface and lateral mixing of NBL waters and inflowing waters, and taking into account the diffusive flux between the NBL and water column above in stratified conditions, the average oxygen consumption rates in summer (in stratified conditions) was about 1.7 mmol $O_2$ m$^{-2}$ h$^{-1}$. Thus, the substantial oxygen depletion in late summer-early autumn 2018 in the near-bottom layer, separated from the rest of the water column by a vertical salinity gradient, was created early and existed during the whole summer-autumn. The observed increase in phosphate concentrations in the hypoxic near-bottom layer in summer 2018 suggests a significant sediment phosphorus release up to 13.5 µmol m$^{-2}$ h$^{-1}$. Intensification of eutrophication effects, including the observed deepening of hypoxia and increased near-bottom phosphate concentrations, could be partly associated with prolonged stratified seasons in the recent decade. We conclude, if similar meteorological conditions as in 2018 could occur more frequently, such extensive hypoxia would be more common in the future in the Gulf of Riga and other coastal basins with similar morphology and human-induced elevated input of nutrients.

**Data availability**

Historical and forcing data can be found in databases (see Sect.2). CTD data are available via SeaDataNet. Buoy profiler data are available upon request.

**Author contribution**

S-TS was the main responsible person for developing methods, analyzing data, and writing the manuscript. UL and JL contributed to developing methods and writing the manuscript. TL contributed by analyzing the data and reviewing the manuscript. MS provided river inflow and Latvian CTD data and contributed to reviewing the manuscript. OS contributed by collecting and analyzing CTD data. IL contributed by analyzing the nutrient data and reviewing the manuscript.



**Competing interests**

The authors declare that they have no conflict of interest.

**Acknowledgements**

5    We thank the agencies and institutes funding and implementing the marine environmental monitoring programs in Estonia and Latvia. Data were provided through the Estonian environmental monitoring information system (KESE), Latvian environmental monitoring database (Latvian Environmental, Geology and Meteorology Center), HELCOM/ICES database, SeaDataNet Pan-European infrastructure for ocean and marine data management, and Copernicus Climate Change Service information. We are thankful to the crew of RV Salme and colleagues who participated in the cruises and data exploration (Ilja

10   Maljutenko for the help with ERA5 data). This work was supported by the Estonian Ministry of Education institutional research funding (IUT19-6), the Estonian Research Council grant (PRG602) and the joint Baltic Sea research and development program (Art 185) through Grant 03F0773A (BONUS INTEGRAL).



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
