# Peer review of "Causes of the extensive hypoxia in the Gulf of Riga in 2018"

_Biogeosciences, 2021_

## Author Response (AR1)

**Relevant changes made in the manuscript**

As per the reviewer comments we restructured our manuscript to follow our hypotheses and study questions.

We shortened the introduction section and removed unnecessary general information, and added hypotheses and study questions in the last paragraph.

We shortened the materials and methods section, and clarified the description of the consumption estimation method.

The results section was substantially reduced and restructured. The new results section includes the analysis of years 2012-2018, as opposed to the former elaborate description of mainly 2018. Still, the focus remains on 2018.

The new layout of the results section consists of the following sub-sections:

3.1. Inter-annual variability of dissolved oxygen in the near-bottom layer

3.2. Seasonal and inter-annual variability in the vertical distribution

3.3. Temporal development of hypoxia in 2018

3.4. Analysis of meteorological and hydrological conditions

3.5. Estimates of oxygen consumption and sediment release of phosphates

In the revised manuscript, we did not include the autonomous profiler results, and also, we left out the year 2018 special survey results.

The discussion section is rewritten based on the formulated hypotheses and study questions.

As recommended, we ordered proofreading for the revised manuscript.

**Reviewer1 comments and answers (given in blue)**

Description of changes made in the manuscript is denoted with yellow.

**General comments:**

This manuscript investigates the underlying causes of extensive hypoxia in 2018 in the Gulf of Riga, using data from the regular monitoring program, a continuous profiling system at the deep part of the basin and a specific survey in September 2018. The authors conclude that the 2018 hypoxia was caused by a combination of several factors: 1) High freshwater and nutrient inputs in autumn 2017 and January 2018 promoting high productivity in the system, 2) inflow of saline waters from the EGB early in the year, due to unusual wind patterns, forming a deep located halocline (deeper than normal resulting in a smaller water volume in the NBL) that was maintained by north-easterly winds during summer, 3) rapid warming of the surface layer strengthening thermal stratification, and 4) reduced ventilation of the NBL during summer. These conclusions are not surprising and basically confirmatory to our present understanding of processes governing hypoxia.

As such, I concur with the authors in their conclusions, but it really surprised me that it takes 32 pages to underpin these conclusions. Unfortunately, the manuscript does not keep a clear stringent structure, as it is filled with repetitions and unnecessary details not used for supporting the conclusions. For example, the results section presents many and elaborate analyses (17 pages) without it being clear how these results support the conclusions. Reading the results section felt more like reading a WQ status report from an environmental agency, where all data have to be presented – relevant or not. The weak point is that the results section does not guide the reader towards the main conclusions!

As I see it, the main problem is that the manuscript is not structured around clearly formulated hypotheses that are subsequently investigated in detail. The authors state that the objective is 'to evaluate the possible role of different forcing factors leading to the observed hypoxia'. I would strongly recommend that all the possible factors/explanations are outlined in detail with appropriate referencing to other literature studies in the introduction and that the M&M section describes how each of these hypotheses will be investigated with rigorous data analyses to address each of them separately. The results section should present only the analyses relevant to the hypotheses and finally, the discussion should centered around relevant scientific discussion points instead of repeating the results. This will require substantial rewriting, but I am also confident that the outcome will be more appealing to the readership of Biogeosciences. I estimate that the main text could be reduced by half.

**Response:** Thank you very much for the comments! We will revise the manuscript according to your comments. The main questions of the study were: 1) What was the reason for the observed extensive near-bottom hypoxia in the Gulf of Riga in 2018? and 2) Was it an exceptional event, or is it a feature that could occur in the Gulf of Riga and similar basins regularly and/or even more often in the future? We did not have clear hypotheses when analysing the measurement data. Of course, it could be suggested that the observed extended hypoxia was probably related to some specific meteorological/hydrographic conditions. The enhanced input of nutrients is just in the background, and it alone does not create sudden hypoxia/anoxia in late summer/autumn. Strong stratification is something supporting hypoxia development. However, for the year 2018, we found that the most important was haline stratification in the deep layer already in spring. The deep stratification was maintained by the inflow of saltier waters from the open Baltic. It should also transport oxygen to the deep layer. However, we estimated that the consumption is higher than advection and diffusion that

resulted in extended hypoxia. By analysing historical data, we showed that hypoxia is occurring more often. Extrapolating this finding and analysing meteorological data in 2018 in relation to the long-term averages, we could conclude that probably such hypoxia will occur in the future even more often. This is the paper in short. We will revise the text in the Results section focusing on these two questions and only presenting the data and analysis supporting the conclusions. Accordingly, we will revise the Introduction and Discussion sections. The text will be shortened. We hope it makes the presentation clearer.

*We revised the text keeping in mind our study questions and hypotheses, and removing unnecessary general information, and shortening the results part.*

In my reading I also found several unclear sentences and sentences that could be sharpened. I have listed some of these under my technical comments, but I stopped commenting on the language after the introduction, realizing that a major rewrite would be required. I do recommend that the next version of the manuscript is proofread by a native English speaker.

**Response:** Thank you. We will follow your suggestions, and when the manuscript is revised, we will order proofreading by a native English speaker.

*We ordered proofreading.*

**Specific comments:**

The introduction is quite long and contains very general, and occasionally trivial, information about processes related to hypoxia (almost textbook like). This information could/should definitely be shortened to present only the most relevant information that leads to the formulation of the objectives and research questions. I suggest that the authors outline all the possible causes underlying hypoxia in the Gulf of Riga, leading to the formulation of specific testable hypotheses. Moreover, I do not think the introduction presents a stronger motivation for the study. I hope the author can present information that explains why 2018 is particularly interesting and why it is relevant to consider a single year. Are years like 2018 expected to be more frequent under current climate change scenarios – i.e. are we expecting more such events to occur in the future? Did 2018 have any ecological consequences such as fish kills, loss of benthic fauna, etc. In summary, the introduction needs to be terser.

**Response:** We will revise and shorten the Introduction section. We will skip the trivial sentences and keep the most relevant text to two main scientific questions (mentioned above). We did not know about the presence of deep haline stratification to hypothesize its importance. However, with the present knowledge, the presentation of the study questions, suggestions, methods and results can be improved (as suggested).

There were no specific ecological consequences. However, the release of phosphates from the sediments is triggered by such hypoxia and counteracts the external nutrient load reduction.

*We revised and shortened the introduction section.*

It would improve the readability of the materials and methods section, if the different data analyses were more clearly linked with specific hypotheses stated in the introduction.

**Response:** We will revise this section and try to relate the methods to the questions.

*We shortened the materials and methods section.*

The calculation of oxygen consumption rates are based on simple box model assumptions, but these calculations are also very sensitive to small differences in salinity bw stations 114 and G1 (the divisor in Eq. 1). The authors should comment on this and how a small potential bias in using values at stations 114 to characterise the inflow of saltier EGB water could influence the calculated rates for oxygen and phosphorus.

**Response:** Yes, we agree. We were able to estimate the uncertainties regarding the variability at station G1 (central Gulf of Riga). The same applies to profiles at station 114 (Irbe Strait). However, the measurements at station 114 could miss the inflowing water mass. Thus, an additional uncertainty factor is present. We will discuss this point in the revised text.

*After careful consideration, we decided to not discuss it in the revised manuscript. We understand the potential uncertainties, especially if the measurements at station 114 did not catch the inflowing waters. The estimates agreed with the results from literature and we applied the method only for the periods with clear near-bottom inflows. Adding this discussion would blur the focus.*

On page 8, it is described that a decline in oxygen concentration should be expected when physical processes are taken into consideration. Does this mean that the authors discarded observations that did not exhibit a decline in oxygen after adjusting for physical processes? If yes, this would bias oxygen consumption rates to higher values, as negative values can be expected by shear randomness. This needs to be clarified.

**Response:** Physical processes (advection and diffusion) should cause an increase in near-bottom layer oxygen concentration. We estimated the expected oxygen concentration and assigned the difference in the expected oxygen concentration and the observed oxygen concentration to oxygen consumption. We did not find negative consumption values based on the measurements (profiles) with a time step of 1-1.5 months.

[Figure]
n/a

On page 9, trends in oxygen and phosphate concentrations are investigated, but why are the authors interested in trends? What do they expect? This is one of many examples, where the formulation of a hypothesis would improve the storyline. Are the authors expecting that expanding hypoxia in the EGB will have an effect on the Gulf of Riga and increase the likelihood of spilling over?

**Response:** We will explain it better. The trends were analysed to determine whether the hypoxic conditions and high phosphate concentrations have occurred more often recently. It answers the question of whether hypoxia observed in 2018 agrees with the long-term changes. In addition, the

finding is that the trends indicate worsening of conditions despite no increase in external nutrient load. Thus, we could conclude that meteorological/hydrographic conditions, e.g., prolonged stratified season, should be responsible.

*We formed hypotheses and revised the text according to them.*

The first paragraph of Section 3.1.2 (Page 10) presents changes over time in the physical parameters in the Ruhnu Deep. From reading, it is not clear why all this information (and with the high level of detail) is presented. Parts of the paragraph are trivial and the text could easily be reduced substantially (e.g. the two first sentences could be removed).

**Response:** We will shorten the text keeping only the relevant information (regarding the main questions).

*We shortened the text keeping only the relevant information.*

On page 12 first paragraph, many numbers are presented, but why are these numbers relevant for the storyline.

**Response:** We will shorten the text keeping only the relevant information (regarding the main questions).

*We shortened the text keeping only the relevant information.*

On page 12 second paragraph, the authors assess the uncertainty of the areal estimates of hypoxia by looking at the distribution of the depth of the hypoxia threshold value. Since the authors have many profiles that are spatially distributed, why didn't they investigate the spatial distribution of the threshold value to see if the depths are horizontally 'constant' over the domain? This would be a more meaningful analysis. Moreover, it is not clear how the authors will use their uncertainty estimate! What is the purpose of this calculation, if it is not used for substantiating the arguments later?

**Response:** We will skip this part. It is relevant to evaluate the uncertainty of the method (when just one profile in the central gulf is used to determine the hypoxic area extent). However, we agree that it is not important for this paper.

*We skipped this part.*

Section 3.2.1: why are the authors presenting all this information on wind patterns? It would be easier to read if the authors formulated a hypothesis about which wind patterns promote hypoxia and then investigate these.

**Response:** We will revise and shorten the text linking the analysis to the suggestion that north-easterly winds could create the inflows of saltier waters (and thus, deep layer stratification in the gulf) and low wind speed and high irradiance/air temperature could support vertical stratification and hinder vertical mixing.

*We revised and shortened the text according to our response above.*

Section 3.2.2: Again, explain why these data are interesting! River discharge data should be presented in a more hypothesis-driven context. Describe the expectations for the data and underpin with analyses.

**Response:** We will revise and shorten the text linking the analysis to the suggestion that large river discharge could bring more nutrients and organic matter to the gulf.

*We revised and shortened the text according to our response above.*

Section 3.2.3: Same comment.

**Response:** See answer regarding Section 3.2.1.

*We revised and shortened the text according to our response above.*

Section 3.3.2: The authors use 1.5 page of text to describe different profiles. This section is longwinded and should be shortened. As an example on page 18 (L. 5- 13), an entire paragraph is used to explain that stratification was stronger in 2018 than in 2017. This could be said with a single sentence. The whole section could easily be reduced to less than half size.

**Response:** We agree and will revise this entire section.

*We revised and shortened the text.*

The discussion is primarily a repetition of the introduction and results sections, and it doesn't read like a discussion section. It is important that the authors bring up pertinent research questions and treat these from the angle: - what do we know, what has this study shown and what can we learn? This approach to the discussion would also highlight the novelty of the study. I believe a rewriting of the discussion to follow the general style of a discussion would be needed.

**Response:** We will revise the manuscript, including the Discussion section as described in our response to the general comment. We will focus on two main questions. Discussion section will be organized around main suggestions and findings by discussing them in relation to the local Gulf of Riga conditions and similar coastal bays.

*We revised the discussion section focusing on the study questions.*

Conclusion: Is there really a need for a concluding section? This section is basically a summary and not a terse concluding paragraph. It needs to be shorter and highlighting the novelties of the study. If this cannot be done, then it is not needed.

**Response:** We think a short concluding section is needed, but it could be shorter by listing the main conclusions and presenting them as the Discussion section's last paragraph.

*We added a short concluding paragraph in the end of the discussion*

**Technical comments:**

*Changes suggested in the specific comments were taken into account and used in the revision process, as per the answers below each comment. Although, some changes per the comment-response pairs are not present in the revised manuscript due to the substantial revision and shortening of the text.*

Page 1, L. 12: How can something be both 'occasionally' and 'dominating'? Wouldn't it be more meaningful to just write 'due to unusual north-easterly winds'.

**Response:** We will rephrase it. North-easterly winds are not unusual, but their dominance is.

Page. 1, L. 14: What do you mean by 'existing stratification'? Existing relative to what!

**Response:** Just "stratification" or "stratification stronger than usual".

Page 1, L. 17: 'prolonged seasonal thermocline and stronger haline stratification'.

**Response:** Agree.

Page 1, L. 19: Should be 'under hypoxic conditions'.

**Response:** Agree.

Page 1, L. 20-22: This last sentence is an exact copy of the last sentence in the conclusion.

**Response:** Concluding remarks will be revised.

Page 2, L. 12: Insert 'a' before permanent halocline. Moreover, I think the authors need to mention that MBIs only give a short-term relief to hypoxia, but on the long term enhance stratification and thereby reduce vertical oxygen transport (Conley et al. 2002; Carstensen et al. 2014).

**Response:** Agree. We will insert this.

Page 2, L. 14-15: This statement is only valid for the eastern Gulf of Finland, I believe. Change to 'In the eastern Gulf of Finland , the south-westerly wind forcing can change the dominant estuarine circulation pattern, leading to erosion of the halocline in the cold season ....'.

**Response:** It is valid almost in the entire Gulf of Finland (e.g., see Lips, U., Laanemets, J., Lips, I., Liblik, T., Suhhova, I. and Suursaar, Ü.: Wind-driven residual circulation and related oxygen and nutrient dynamics in the Gulf of Finland (Baltic Sea) in winter, Estuar. Coast. Shelf Sci., 195, 4–15, 2017).

Page 2, L. 17: 'where the halocline is absent and a seasonal thermocline restricts vertical mixing, promoting hypoxia in the near-bottom layer and sediment phosphorus release in late summer-autumn (refs.).'

**Response:** Thank you.

Page 2, L. 21: Replace with 'Sedimentation of organic matter, stimulated by nutrient inputs, can cause severe oxygen deficiency under specific meteorological/hydrographic conditions, .....'.

**Response:** Thank you.

Page 2, L. 29: 'Water and salt budgets of the gulf are governed by the river discharge, precipitation-evaporation balance and water exchange ...'

**Response:** Thank you.

Page 2, L. 31: Insert 'surface' before precipitation.

**Response:** Thank you.

Page 2, L. 34: '… of the gulf with the Daugava River contributing about ….'.

**Response:** Thank you.

Page 3, L. 1-2: This sentence is unclear, please rewrite. Also, insert 'to' before 'about three years'.

**Response:** Thank you. We will try to make it clearer. We just wanted to say that Lilover et al. (1998) estimated that the gulf's water would be renewed about every three years.

Page 3, L. 6: Replace with 'while these hydrographical features are 5 m and 0,04 km2, respectively, for the Suur Strait'.

Response: Thank you. We will rephrase it as recommended, taking also into account Reviewer2 suggestions.

Page 3, L. 21: 'the whole water column is well mixed in winter.' The homogenous distribution is implicit.

**Response:** Thank you. We will remove the implicit part of the sentence.

Page 3, L. 23: Remove 'the' before 'strongest in August'.

**Response:** Thank you. We will remove 'the'.

Page 3, L. 25: Delete 'of the water column' – no need to specify this. '… occurred in years with the highest summer surface temperature and spring river discharge'.

**Response:** Thank you. We will delete 'of the water column'.

Page 3, L. 26: Place comma after '(2017)'.

**Response:** OK.

Page 3, L. 32-33: I suggest to use 'spring' consistently instead of intermixing 'spring' and 'vernal'.

**Response:** OK, we will do so throughout the manuscript.

Page 4, L. 8: Replace 'during the last decades' with 'in recent decades'. It should also be mentioned that the halocline position has shifted upwards in the water column, enabling denser and oxygen-depleted waters to spill over into the Gulf of Riga (cf. Carstensen et al. 2014).

**Response:** We will change it as suggested. We consider to refer to Carstensen et al. (2014), although we do not think upward shifting of the halocline (which is at the depths of 60-70 m) is directly linked to the inflow of less oxygenated waters over the sill with a depth of 20 m.

Page 3, L. 10: Be more specific! Which northern Baltic coastal basin are you talking about? Place comma after 'nutrient input'.

**Response:** We guess it refers to the text on page 4. We will add the study site name to the Jokinen et al. (2018) reference. It is Haverö – a small and enclosed basin in the middle of the Archipelago Sea.

Page 3, L. 11-12: What does this sentence refer to? Is it needed? Delete?

**Response:** We guess it refers to the text on page 4. We will rephrase it to make it clear that the authors (Jokinen et al. 2018) explained the found more frequent occurrence of hypoxia in their study site (Höver) already since the early 1900s due to shoaling of the basin, meaning, much earlier than the nutrients loads did increase in the second half of the 20th century.

Page 3, L. 14: Are the inputs in numbers really needed when all you want to say is that nutrient inputs are currently higher than required by the BSAP.

**Response:** Thank you. We will consider shortening it.

Page 4, L. 17-18: Replace with 'following by stagnation'.

**Response:** Thank you.

Page 4, L. 18: 'Since riverine phosphorus input is <15% compared to phosphorus pool in the water column (Yurkovskis, 2004), ......'

**Response:** Thank you. We will rephrase it as suggested.

Page 4, L. 21: 'For instance, phosphorus release in the order of …..'.

**Response:** Thank you. We will rewrite it as suggested.

Page 4, L. 22: Replace 'poor' with 'low'. Suggestion: 'that counteracts efforts to reduce phosphorus inputs to the gulf.'

**Response:** Thank you. We will replace 'poor' with 'low' and rewrite as suggested.

Page 4, L. 29: 'we estimated oxygen consumption and sediment phosphorus release rates under the observed hypoxic conditions.'

**Response:** Thank you. We will rephrase it.

Page 7, L. 11: Insert 'defining' before hypoxia.

**Response:** OK.

Page 7, L. 14: This in under the assumption of horizontal homogeneity and this should be specified.

**Response:** We will skip the special survey part regarding the areal extent of hypoxia.

Page 7, L. 21: Explain how close to the bottom the profiles get!

**Response:** We will add this information to the first paragraph of the Method section. The profiles covered in most cases the depth range from 2 to 52 m. The sea depth at station G1 is 54 m.

Page 9, L. 8: 'low sampling frequency' or 'scarcity of data'.

**Response:** Thank you.

Page 12, L. 10: 'standard error' should be 'standard deviation'.

**Response:** OK.

Fig. 4: I suggest to show (a) as a cumulative wind transport for the different years invidually. Wind transports for the years 2005-2017 could be shown with a thin line and then 2018 could be highlighted.

**Response:** Thank you for the suggestion. We will consider making a cumulative wind transport figure presenting individual years, although we think cumulative wind stress could be more representative for the straits.

Page 16, first paragraph: Are all the details here really needed for making the point? Again, it would be better if these assessments were done in relation with a hypothesis. One important question to ask, which the authors haven't done, is whether the bottom water inflows from the EGB are mixed with surface water when passing across the sill, implying that the mechanism is not due to oxygen-depleted water spilling in, because the inflow gets 'oxygenated'. This would point to that respiration within the Gulf of Riga is the important process consuming oxygen, i.e. it is not imported hypoxia.

**Response:** We will condense the text. However, we suggest that hypoxia is developed locally. We do not have any evidence that it could be transported from the Baltic Proper. The sill depth is too shallow, and all measured oxygen concentrations in the Irbe Strait near-bottom layer are far from hypoxia.

Page 21, L. 4-6: Is this relevant for the study?

**Response:** We will shorten this part. However, we find that the observed rapid increase of the UML depth is relevant for further discussion about the characteristics of inflowing water.

Page 21, L. 12-15: This is another question about monitoring frequencies, but essentially it is not part of this study. So, why bring up that discussion here?

**Response:** What we try to emphasize here is the uncertainty of the single measurements versus continuous monitoring. We would like to keep it in the text but move it to the Discussion section.

Page 22, L. 3-4: This sentence just repeats what was stated in the materials and methods section.

**Response:** We will remove it.

Page 22, L. 17-18: How do the authors know that outflow prevailed during this period?

**Response:** This assumption was based on CTD data from the August monitoring cruise (Fig. 7), showing an outflow in the deeper layer near Irbe Strait (station 114), and the lack of inflow favoring winds (Fig. 4; monthly mean winds from August to October were from SW direction).

Table 1+2: These data are much better displayed as a figure, showing budgets for salt, oxygen and phosphate, i.e. three budget figures with all three variables in the same plot.

**Response:** Thank you. We will make figures and consider adding them instead of the tables or restructuring them (including fewer rows).

Page 23, L. 9-11: Repetition from materials and methods.

**Response:** We will remove the repetition.

Page 30, L. 12: This is not always the case! When the spring bloom sediments, diatoms are often viable and can continue living on the sediment surface for months. Furthermore, the relatively low water temperatures in spring will reduce respiration processes. Thus, it is generally believed that there will be a delayed response between spring bloom sedimentation and respiration.

**Response:** Thank you. We will discuss it and add relevant references. However, since the near-bottom temperature does not change much during the summer months, also the respiration process does not change much in time.

Page 31, L. 15-16: Was this because the pool of Fe-bound phosphate was emptied? Worth considering.

**Response:** Thank you. Yes, it could be the case. We will consider this and add relevant reference(s).

Page 32, L. 20-22: This sentence is an exact copy of the last sentence of the abstract.

**Response:** As mentioned above, we will rephrase the concluding remarks.

**Reviewer2 comments and answers (given in blue)**

Description of changes made in the manuscript is denoted with yellow.

**General comments**

*1. The manuscript presents a case of hypoxia developed in the Gulf of Riga in 2018 and analyses its causes determined mainly by reduced ventilation of the deep layers due to hydrophysics. As such, the mechanism of hypoxia emergence as a result of imbalance between oxygen biological demand and its supply in some water and sediments domains is known for decades. Neither the seasonal occurrence of hypoxia is exceptional in the Gulf of Riga. Therefore, such regional case study of a particular year might be interesting for the wide global audience only if it would present something not only geographically but also methodologically new and generalized. Besides, the manuscript in its current state appears as a technical report to some monitoring agency rather than precise and focused scientific paper. In that I concur with many comments and suggestions already made by Reviewer #1.*

Response: Thank you very much for the comments. We will revise the manuscript taking into account this general comment as well as referred comments by Reviewer1. The parts not so relevant to the main questions of the study will be shortened, keeping only those sections/paragraphs that support discussion on the main results. As also replied to Revier1, we have two main questions: 1) What was the reason for the observed extensive near-bottom hypoxia in the Gulf of Riga in 2018? and 2) Was it an exceptional event, or is it a feature that could occur in the Gulf of Riga and similar basins regularly and/or even more often in the future? We hope the results focussed on these questions are of interest to the wider audience.

*We revised the text keeping in mind our study questions and hypotheses, and removing unnecessary general information, and shortening the results part.*

*2. Selecting for analysis only 2017 and 2018 without clear explanation of the choice, you pretty much reduce the interest in the manuscript for global audience and even make questionable its suitability for Biogeosciences vs. some other journals, explicitly dealing with either the Baltic Sea problems or regional issues. In that respect, the manuscript can be saved by a comparative analysis involving a set of "hypoxia years" (e.g. 1996, 2003, 2008, 2012, 2014, 2015, 2018 (some of them you list at p.12, lines 1-5) and a set of "hypoxia-free years" of your choice. If you you'll find the similarities or even regularities between the cases, such causative, mainly geophysical (?) relationships, even if semi-quantitative, could then be used together with available and evolving climate projections of relevant parameters. If there are none found, such "negative" result, being clearly shown and explained, would still increase the scientific knowledge about the Gulf of Riga.*

Response: We agree this is an important point and will include relevant text in the Discussion section. However, we cannot compare the long-term data sets since we have profile data available from 2012. Before that, the rare near-bottom oxygen measurements were done, which are analysed and the relevant analysis is included (as trend estimates since 2005). Thus, we are not able to include any statistical analysis regarding stratification, inflow-outflow estimates, etc. Instead, we will improve qualitative discussion on this matter.

*We expanded our analysis and included the years, from when we had consistent CTD data, 2012-2018.*

*3. The occurrence of hypoxia (O2 less than 2 mL/L) in the Gulf of Riga is not new and was often observed already in the 1970s and 1980s (see, for instance, http://nest.su.se/nest/ and then go to Baltic Sea=>Marine distributed databases), when also deep-layer salinity was higher until its drop in about 1990 and have been fluctuated without a long-term trend since then. Since it remains unclear why your data analysis was limited to 2005-2018 (p. 6, line 30), a short text expanding a time perspective would help to set a scene for the global audience.*

Response: The earlier conclusions based on similar monitoring data are included. It is mentioned in the Introduction section, why we analyse these data sets, but possibly hidden a bit. When revising the manuscript, we will keep only the text relevant to the main focus in the Introduction. We hope it will make it clearer.

*We revised the introduction section.*

*4. Both the text style (the very manner of describing and presenting, e.g. describing in detail what is seen in Figures, especially features that would not be used further) and its volume (could, perhaps, be halved) look inappropriate to me for the scientific paper. In addition to and supporting suggestions by Reviewer #1, pieces of text that could and should be condensed are indicated in Specific comments, even with a suggested example of the editing.*

Response: We will improve the manuscript taking into account this comment and suggestions (as well the similar suggestion by Reviewer1) regarding the presentation style and focus of the manuscript.

*We shortened the text keeping only the relevant information. And, we formed hypotheses and revised the text according to them.*

*Specific comments*

*Changes suggested in the specific comments were taken into account and used in the revision process, as per the answers below each comment. Although, some changes per the comment-response pairs are not present in the revised manuscript due to the substantial revision and shortening of the text.*

*p.1, line 10 – is it really exceptional, in what sense and by which characteristics – minimum of absolute or % saturation oxygen concentration, extent of hypoxic zone?*

Response: The extent of hypoxia was the largest in 2018 compared with the other years since 2012. However, we agree that it is better to call it "extensive". It also fits better to our suggestion that it is nothing exceptional but a development that will occur in the future if the load is not reduced and the meteorological conditions support longer stratified periods.

*p.1, line 11 – "Forcing data…" appears as a kind of slang, should be explained, something like "meteorological" or "weather", especially in Abstract. Forcing of what, how, etc... For instance, could temperature and salinity per se be considered as forcing for oxygen because of the oxygen saturation?*

Response: We will replace "forcing" with "meteorological", to be precise.

*p.2, lines 3-7 – sloppy unnecessary description, remember about long nutrient residence times and the vicious circle; what about point sources with undertreated discharges from WWTPs? Can be easily removed altogether.*

Response: We will shorten this section avoiding unnecessary sloppy or straightforward descriptions.

*p.2, line 14 – among hypoxia suddenly about aeration, as, for instance, if there were no hypoxia events in the Gulf of Finland; add and mention Lehtoranta et al. doi: 10.1016/j.jmarsys.2017.02.001, refer to Stoicescu et al., 2019.*

Response: We will add suggested references.

*p.3, lines 4-5 – could be modified: "The Gulf of Riga water exchange with the Baltic Proper takes place via the Irbe Strait in the west and the Suur Strait in the north (Petrov, 1979; Astok et al., 1999) with dominating contribution of the Irbe Strait (Lips et al., 1995; Skudra and Lips, (2017)."*

Response: Thank you, it is better.

*p. 3, lines 6-19 – Can then be compressed down to two ideas – general surface and deep-water patterns, and its seasonal alterations. Should it be placed in Introduction vs. Discussion?*

Response: We will keep this in the Introduction, but in a shortened version.

*p.3, lines 21-28 – the amount of text can be halved by deleting trivial things, some of which were already indicated above. Just as an example: "In winter, the whole water column is well mixed. In summer, stratification is mainly maintained by the seasonal thermocline, while the contribution of haline stratification is rather moderate (Stipa et al., 1999; Liblik et al., 2017). (opt. - Summer CTD profiles from 1993–2012 have shown that) In 1993-2012, the strongest stratification of water column occurred in the years with the highest upper layer temperature in summer and river runoff in spring (Skudra and Lips, 2017)."*

Response: Thank you. Yes, we agree and follow your suggestion.

*p.3, line 30 – Another example of necessary cut-out is a trivial description of the annual cycle of DO. It would be enough to have nice color picture(s) with isopleths further, in the Results or even Discussion*

Response: We will shorten the text.

*p.4, lines 14-23 – where and how information on loads would be used further; if retained during revision, reference to (Yurkovskis, 2004) about external input vs. internal processes could be extended with references to Savchuk (2002, 2005, 2018), where such things are discussed in detail. Although all this discussion about the nutrient buffer capacity as well as mentioning of DIP vs. Ox fluxes should be transferred to Discussion, if relevant. Reference to the HELCOM Periodic Load Compilations, both*

*published and publicly available as time series at http://nest.su.se/helcom_plc could be appropriate here.*

Response: Thank you. We will use load data from PLC (available until 2017; thus, relevant for our analysis) in the analysis. References and text will appear mostly in the Discussion, as suggested.

*To resume, the entire Introduction must be thoroughly re-written and condensed, replacing the textbook-like general geographical and imprecise descriptions and numbers, which would not be used further either in Results or in Discussion, by indication of why you made this study, i.e. what the problem is and how you dealt with it. The questions and hypothesis should be clearly formulated to be then positively or negatively answered in Discussion. Evidently, all that have to be made for an analysis of extended set of "hypoxic years".*

Response: As explained above, we will revise the Introduction section, condense and focus on two main questions: 1) What was the reason for the observed extensive near-bottom hypoxia in the Gulf of Riga in 2018? and 2) Was it an exceptional event, or is it a feature that could occur in the Gulf of Riga and similar basins regularly and/or even more often in the future? Unnecessary paragraphs or sentences will be deleted or shortened.

*Section 2. Do we really need such detailed description of equipment? Could the data sources be just moved to the Acknowledgments in the end?*

Response: We think the details are necessary. However, we will condense the text.

*Section 3.1.1 must be drastically condensed and replaced instead with O2 vs. Salinity graphs and regressions for the time interval expanded backwards.*

Response: We would prefer to keep this part as it is. Almost no monitoring data is available before 2005, except in the mid-1990s. Thus, we cannot extend the trend analysis.

*Section 3.1.2 must be condensed by removing boring description of nice Fig. 3, which, however, have to be extended back in time. Numerical values should be collected in Table, but only if you would use them further in Results or Discussion. For instance, a too verbose description of the hypoxia extent in 2018 (p. 12, lines 6-12) could be quite condensed just indicating that the values from the survey justify estimates based on the central stations, which are given in the previous paragraph (and compiled in Table).*

Response: We agree that the text can be shortened (will do so). However, Fig. 3 cannot be extended since we do not have dissolved oxygen profile data before 2012.

*Section 3.2. The time interval should be unified and graph's description must be condensed by editing the boring description of what is seen on graphs, reformulating and stressing those features and peculiarities important for further Discussion.*

Response: The time interval is the same for all graphs (1979-2018) except the river runoff. For the latter, we have data since 1993. The idea is to show the year 2018 in comparison with the long-term averages and variability. We would prefer to keep the graphs as they are. However, we will shorten the text by focusing on those aspects necessary/important for explaining and discussing the results.

*Section 3.3. have to be entirely and much more laconically re-written (avoiding detailed description of pictures and stressing only the features used in further analysis and discussion) according to the analysis of expanded set of hypoxic and non-hypoxic years, suggested above.*

Response: We agree and will shorten the text part accordingly.

*Section 3.4 could be retaining as an example obtained with a specific equipment, but must be very much condensed by some generalizing instead of describing in detail what has happened from day to day. Besides, explain and justify, why the uncertainty estimated for the synoptic scale could give an estimate for two-week period and would be higher at a seasonal one-two months scale (p. 21, lines 12-15).*

Response: We will shorten the text focusing only on the near-bottom layer and the sudden deepening of the upper mixed layer relevant to discussing the results. Regarding the uncertainty, we suggest that if a single measured oxygen value is taken as a characteristic value for a period, then the uncertainty is higher for a longer period.

*Section 3.5 – It should present estimates obtained from a "hypoxic years" vs. "non-hypoxic years" sets. Numbers for calculations should be moved in Appendix, if necessary at all.*

Response: We cannot provide estimates for the years without dissolved oxygen profiles (before 2012). Also, the estimates could be biased when the inflow through the Irbe Strait is not evident in the deeper layer at station 114. We will restructure the table keeping only necessary rows, and provide some estimates for the other years (where possible).

*Evidently, the entire Discussion must be re-written according to results from "hypoxic years" analysis. It could also be enriched by considering also results and conclusions for the geographical locations other than the Gulf of Riga and the Baltic Sea.*

Response: We will revise the Discussion section taking into account similar comments from you and Reviewer1. We focus the discussion on the mentioned two main questions and try to enlarge the geographical relevance by referring to scientific publications from elsewhere..

*Technical suggestions and corrections (because of expected re-writing, below I suggest a few corrections to the pieces that could likely be retained)*

*p.2, line 10 – Conley et al., 2009 should be added here as well*

Response: We agree.

*p.2, line 12 – be consistent, use the Baltic Proper everywhere, or, if necessary, use names of narrower localities – the Gotland Deep, the Bornholm Deep, the Eastern Gotland basin, etc.*

Response: We will re-check it, although, in this particular case, we think the name is correct and understandable also for readers from other regions.

*p.3, line 1 – "...being in annual balance " would be easier reading*

Response: We agree.

*p.3, line 2 – "...period OF about..."*

Response: We agree.

*p.3, line 5 – refer also to Astok, V., Otsmann, M., Suursaar, Ü., 1999. Water exchange as the main physical process in semi-enclosed marine systems: the Gulf of Riga case. Hydrobiologia 393, 11 –18.*

Response: Thank you. Will do so.

*p.3, line 6 – "…Suur Straight IS OF 5 m2..."*

Response: OK, this part is rephrased.

*p.4, line 14 – If it AT LEVELS then it should be "… about 90.5 and 2.5 thousand tons a year, respectively"*

Response: OK, this part is rephrased.

*… to be continued after the Major Revision*

**Reviewer3 comments and answers:**

Description of changes made in the manuscript is denoted with yellow.

*Comment: The topic of the manuscript is interesting and the analysis of data is very useful. However, the manuscript is too unfocussed and much too long. The reading was difficult for me because a red line does not exist. One possibility would be to only focus on the oxygen budget of the deep water of the Gulf of Riga and to compare the fluxes and inventories from the observations with numerical model data. With the help of a model it would be possible to disentangle the drivers of hypoxia and also to evaluate longer time series. As the natural variability of the system is large, the short study period does not allow to investigate statistically significant trends. Hence, a major revision of the manuscript is needed.*

Answer:

Thank you for your short comment. We will rewrite the manuscript by shortening it and focussing on two main questions: 1) What was the reason for the observed extensive near-bottom hypoxia in the Gulf of Riga in 2018? and 2) Was it an exceptional event or is it a feature that could occur in the Gulf of Riga and similar basins regularly and/or even more often in the future?

We will keep the oxygen consumption estimates based on the introduced rather simple method accounting for fluxes due to physical processes, but would not enlarge this part. We consider that developing and using a numerical model is out of the scope of this manuscript. Unfortunately, the model outcomes that are freely available (e.g., from CMEMS) are not well enough simulating the oxygen dynamics in the Gulf of Riga deeper layers in stratified conditions.

*We revised the text keeping in mind our study questions and hypotheses, and removing unnecessary general information, and shortening the results part.*

---

## Author Response (AR2)

**Oleg Savchuk**

Review of the revised manuscript by Stella-Theresa Stoicescu, J. Laanemets, T. Liblik, M. Skudra, O. Samlas, I. Lips and U. Lips: "Causes of the extensive hypoxia in the Gulf of Riga in 2018".

1. I have intentionally started reading the revised version without looking first either in the extended Responses or "ATC1" file with all the corrections indicated in the text. I noticed some improvements in recommended streamlining of the Introduction with the addition of hypothesis and study questions, which, however, have not converted this local case study into something more attractive for the global audience of the Biogeosciences. Another positive improvement was an expansion analysis over 2012-2018 instead of focusing on 2018 only.

2. Unfortunately, two major faults are left hardly revised. Despite some restructuring, the style and length of Results section is almost unchanged: the same unnecessary lengthy, boringly detailed description of what is already clearly seen in Figures and Tables, which actually looks as "listing" rather than description. As a formal confirmation, the text length of Results became only slightly shorter, reduced from 17 to 14 pages, instead of recommended by all the reviewers, at least, halving it, for presenting to readers only most essential statements and conclusions drawn from these graphs and numbers. Perhaps, the Discussion has also slightly improved, however, again being expanded and diluted with some common places and trivial statements. Unfortunately, it has not been enriched much by considering also results and conclusions about hypoxia at the geographical locations other than the Gulf of Riga and the Baltic Sea.

3. To my mind, such superficial revision of the manuscript almost ignoring suggestions by reviewers, have not increased either scientific significance and quality of the manuscript or its presentation quality – it still looks as a Technical Report for the local funding authorities rather than a scientific "brick" into the knowledge about importance of hypoxia even for a very local Gulf of Riga. Therefore, although I could recommend the further Major revision, I will not revise the next iteration.

*We would like to thank Oleg Savchuk for his comments and suggestions. Although, we do not agree that we 'almost ignored' suggestions by reviewers. We tried to shorten and streamline the Results and Discussion sections once more during the second revision of the manuscript.*

**Anonymous**

This is my second assessment of this manuscript and I believe the manuscript has improved. Nevertheless, there are still some issues that need to be addressed. Essentially, this is a relatively simple study, investigating the causes of the extensive hypoxia in 2018 in the Gulf of Riga. The authors have formulated three guiding research questions, leading to a general hypothesis. Given these clear objectives of the study, the storyline should be rather straightforward. The authors arrive at well-substantiated conclusions, and I agree with these, but I am wondering if 27 pages are really needed for this. Although I can see that the authors have reduced the text since the last submission, I still think that the manuscript could be made terser and more focused.

Whereas introduction, M&M and Sections 3.1 and 3.2 are well written, the manuscript starts loosing focus from section 3.3 onwards. One problem is that the authors start mixing discussion issues into the results section from this point onwards. The authors need to present their results in an objective manner and keep their subjective assessment for the discussion. The discussion should focus on the research questions outlined at the end of the introduction, rather than repeating the results and giving general statements about causes of hypoxia etc. I suggest the authors to introduce subheadings for the discussion listing the key research question being addressed. This will help keep the manuscript focusing on answering the questions from the introduction. In doing so, I believe the text could be reduced considerably.

*We modified the results part (from section 3.3 onwards), keeping only objective descriptions of results. We restructured and rewrote the discussion, keeping in mind the study questions and removed the repetition of results. Although we did not introduce subheadings, we structured the discussion focusing firstly on the first study question (stating that 2018 agreed with the long-term trend, but pointing out that the reasons behind the extent of hypoxia need further elaboration). Then we move on to describe the factors influencing hypoxia, using 2018 as an example year and comparing it briefly with other analyzed years in regard of consumption estimates, meteorological forcing factors etc. (second study question - What were the reasons behind the observed hypoxia?). Next, there is a paragraph about future projections and what these could mean for the Gulf of Riga, plus the speculation on phosphorus trends (third study question - Was it a feature that could occur in the GoR and similar basins regularly and/or even more often in the future?). Finally, a short concluding paragraph was added.*

Another question that I would like the authors to consider is whether the increase in phosphate releases over time is directly linked to increases in hypoxia only, or if there could be an accumulation of phosphorus in the sediment pool as well over this period.

*We added some discussion on this topic (Page 21, l. 1-7).*

Thus, I believe the study is correct and deserves publication, but I would urge the authors to make it more focused on the specific research questions and to make a clear distinction between results and discussion.

**Detailed comments:**

Page 2, l. 5-7: This sentence reads a bit like arm-waving. Please elaborate the mechanistic cause-effect relations between climate change and hypoxia.

*We supplemented this sentence.*

Page 2, l. 9-10: Maybe a better reference is the review by Reusch et al (2018) Sci.Rep.

*We added the suggested reference.*

Page 2, l. 12-13: See also paper by Carstensen & Conley (2019) Limnol.Oceanogr.Bull.

*We added the suggested reference.*

Page 3, l. 3: This sentence reads awkward. Assuming what? Do you mean 'Considering the …'?

*We changed the sentence as per the suggestion.*

Page 3, l. 9: Use either 'deep layer' or 'deep waters', not both. It is verbose.

*Noted and changed.*

Page 3, l. 15: Same comment as above.

*Noted and changed.*

Page 4, l. 16: Suggest to use 'speculate' rather than 'predict'. This is what will be discussed, right?

*We changed the sentence as per the suggestion.*

Figure 1: Pärnu River is marked by white square, should be green as for the other rivers.

*Figure changed.*

Page 6, l. 27: better to write 'assuming an even horizontal depth distribution for the occurrence of hypoxia'.

*We changed the sentence as per the suggestion.*

Page 7, l. 6: What about Pärnu River? Weren't there any discharge data available? From Estonian monitoring authorities?

*In the previous versions we did not include discharge data from Pärnu river, because the data available was not representative of the flow estimates in the river mouth (measurements are taken some 30 km inland from the mouth). But, upon learning that these exact same data are used in the PLC calculations, representing the whole Pärnu river runoff, we have now included these data in our calculations as well. It did not change the presentation and results, but dataset used is more complete now.*

Page 7, l. 26: Extracted from where?

*Added specification.*

Page 7, l. 31: 'We introduced a coarse method for estimating …'

*We changed the sentence as per the suggestion.*

Page 10, l. 7: 'the lower sampling frequency'

*We changed the sentence as per the suggestion.*

Page 10, l. 8: 'since 2012, except for 2016 and 2017'.

*We changed the sentence as per the suggestion.*

Page 11, l. 2: 'did not at ..' Remove 'exist'.

*We changed the sentence as per the suggestion.*

Page 11, l. 13: 'was apparent'

*We changed the sentence as per the suggestion.*

Page 11, l. 15: Do you mean 'The lowest oxygen concentrations were observed …'?

*We changed the sentence as per the suggestion.*

Page 11, l. 17: Replace with 'The first occurrence of the seasonal hypoxia was observed in July in 2014, August in 2018, ....'

*We changed the sentence as per the suggestion.*

Page 11, l. 18: Replace 'border' with 'boundary'.

*We changed the sentence as per the suggestion.*

Page 11, l. 23: 'dominated'. Check for consistent use of past tense.

*We changed the sentence as per the suggestion.*

Page 11, l. 27: 'were'. Use past tense consistently.

*We changed the sentence as per the suggestion.*

Page 12, l. 3: Replace 'related to' with 'were associated with'.

*We changed the sentence as per the suggestion.*

Page 13, l. 19: Insert 'previous' before 'year'.

*We changed the sentence as per the suggestion.*

Page 13, l. 21-22: 'in vertical stratification during spring between these two years'.

*We modified the sentence keeping in mind the suggestion.*

Page 19, l. 3: 'during destratification in autumn' rather than 'in the stratification decay in autumn'

*We changed the sentence as per the suggestion.*

Page 19, l. 5-11: These sentences are more discussion than results.

*These sentences were removed from the results.*

Page 19, l. 15-16: Rephrase this sentence moving the verb forward.

*We modified the sentence.*

Page 20, last paragraph: This paragraph reads more like discussion than results. Please keep a clear distinction between results and discussion, also to avoid repetition in the discussion.

*Noted. We removed all discussion-like sentences from the results.*

Page 21, l. 13: 'respiration' is better than 'consumption' here.

*We changed the sentence as per the suggestion.*

Page 24, l. 9: 'If oxygen consumption exceeds oxygen supply ...'

*We changed the sentence as per the suggestion.*

Page 24, l. 28: 'have accelerated recently'.

*We changed the sentence as per the suggestion.*

Page 25, l. 11: What about the degradability of the organic matter. In autumn, much of the labile organic matter has already been degraded and more refractory organic matter remains.

*Aigars et al (2015) suggested, that higher consumption in spring-early summer could be related to the availability of degradable organic material is available (settling of spring bloom, although the impact is smoothed over time). We added word "degradable" into this sentence (Page 19, l. 22).*

Page 25, l. 30: 'summer deoxygenation was higher compared to other years'.

*We changed the sentence as per the suggestion*

Page 25, l. 32: 'uplift of the almost oxygen depleted near-bottom waters'. 'boundary' instead of 'border'

*We changed the sentences as per the suggestion*

Page 25, last paragraph: Isn't this paragraph a repetition of the results?

*Removed.*

Page 26, l. 4: Is this where the discussion really takes off, by addressing the research questions?

*We restructured the discussion, keeping in mind the study questions.*

Page 26, l. 18-19: 'Lower wind speed reduces vertical mixing and enhances stratification'.

*We changed the sentence as per the suggestion.*

Page 26, l. 19-21: What is the main message here? What is the difference between have a more steady (constant) exchange relative to having a more variable?

*Removed*